# Re-TASK: Revisiting LLM Tasks from Capability, Skill, and Knowledge Perspectives

## Abstract

The Chain-of-Thought (CoT) paradigm has become a pivotal method for solving complex problems. However, its application to intricate, domain-specific tasks remains challenging, as large language models (LLMs) often struggle to accurately decompose these tasks and, even when decomposition is correct, fail to execute the subtasks effectively. This paper introduces the **Re-TASK** framework, a novel theoretical model that **Re**visits LLM **T**asks from c**A**pability, **S**kill, and **K**nowledge perspectives, drawing on the principles of Bloom's Taxonomy and Knowledge Space Theory. While CoT offers a workflow perspective on tasks, the Re-TASK framework introduces a Chain-of-Learning view, illustrating how tasks and their corresponding subtasks depend on various capability items. Each capability item is further dissected into its constituent aspects of knowledge and skills. Our framework reveals that many CoT failures in domain-specific tasks stem from insufficient knowledge or inadequate skill adaptation. In response, we combine CoT with the Re-TASK framework and implement a carefully designed **Re-TASK prompting** strategy to improve task performance. Specifically, we identify core capability items linked to tasks and subtasks, then strengthen these capabilities through targeted knowledge injection and skill adaptation. We validate the Re-TASK framework on three datasets across the law, finance, and mathematics domains, achieving significant improvements over the baseline models. Notably, our approach yields a remarkable 44.42% improvement with the Yi-1.5-9B model and a 33.08% improvement with the Llama3-Chinese-8b on the legal dataset. These experimental results confirm the effectiveness of the Re-TASK framework, demonstrating substantial enhancements in both the performance and applicability of LLMs.

## 1 Introduction

As the scale of large language models (LLMs), such as the GPT-4, Claude, and Gemini series (Achiam et al., 2023; Anthropic, 2024; Team et al., 2023), as well as their open-source counterparts like the LLaMA, Mistral, and Qwen series (Touvron et al., 2023; Jiang et al., 2023; Bai et al., 2023), continues to increase, their general capabilities in natural language processing (NLP) tasks have shown substantial improvements. Despite these advancements, however, these models often struggle with complex reasoning tasks, particularly those that are domain-specific. The Chain-of-Thought (CoT) technique (Wei et al., 2022; Kojima et al., 2022; Wang et al., 2023; Zhou et al., 2023) has emerged as a promising paradigm by decomposing complex tasks into a series of subtasks in a divide-and-conquer manner. Yet, the application of CoT to domain-specific tasks faces significant challenges in both task decomposition (Kambhampati, 2024) and subtask execution (Lightman et al., 2024) due to a lack of domain knowledge and specialized capabilities.

The idea that an individual's capabilities directly influence task performance is well supported by educational theories, notably Bloom's Taxonomy (Bloom, 2010) and Knowledge Space Theory (KST) (Doignon & Falmagne, 1985). Bloom's Taxonomy outlines how educational objectives are achieved through structured instructional activities, each involving specific knowledge and cognitive processes. Similarly, KST emphasizes the sequential dependencies between learning items, forming *"learning pathways"* that guide learners from foundational knowledge to mastery.

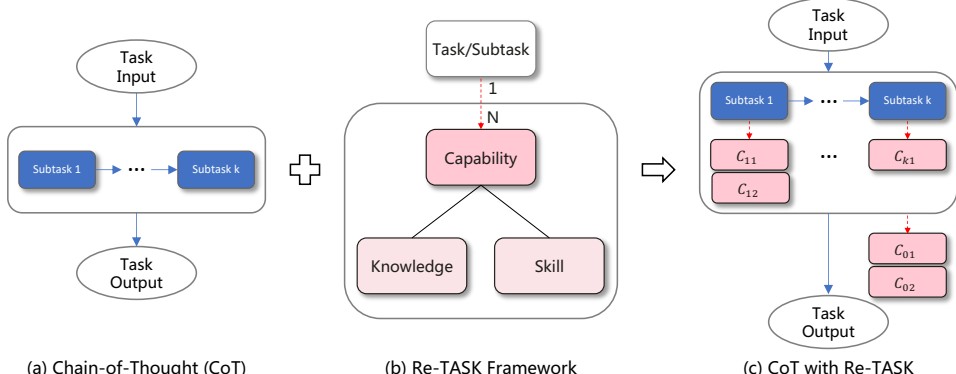

(a) Chain-of-Thought (CoT)  (b) Re-TASK Framework  (c) CoT with Re-TASK

Figure 1: The Chain-of-Thought (CoT) provides a workflow perspective on tasks (blue arrow), while the Re-TASK framework introduces a Chain-of-Learning view (red dashed arrow), demonstrating how tasks and subtasks depend on various $N$ capability items. Combining CoT with Re-TASK enhances CoT's performance in both task decomposition and subtask execution. Here, $C_{ij}$ represents the capability item associated with subtask $i$ (where $i = 1, \cdots, k$), and $C_{0j}$ is associated with the overall task for task decomposition.

Building on these insights into the Chain-of-Learning, we introduce the Re-TASK framework, which revisits LLM tasks through the lenses of capability, skill, and knowledge (see Figure 1(b)). This framework posits that the successful completion of tasks[1] depends on sequentially mastering multiple capability items, with each item further dissected into its constituent aspects of knowledge and skills. The distinction between knowledge acquisition and skill application also aligns with the perspective of Bengio & Hu (2023), who argue that effective reasoning requires both a robust world knowledge model and a powerful inference engine capable of generating solutions consistent with that model.

Our framework posits that failures in the CoT paradigm, particularly in task decomposition and subtask execution, can be attributed to a lack of corresponding capabilities due to either insufficient knowledge or inadequate knowledge-skill adaptation (or skill adaptation for short). First, LLMs may lack relevant knowledge because of limited access to proprietary data or issues related to data timeliness. Second, even when the knowledge is available, LLMs often struggle to effectively apply it to solve complex tasks, leading to suboptimal performance. Techniques such as Retrieval-Augmented Generation (Lewis et al., 2020) can inject knowledge into the context, but models may still underperform due to inadequate skill adaptation on utilizing the retrieved knowledge, ultimately resulting in task failures.

To address these issues, we propose integrating the CoT paradigm with the Re-TASK framework to enhance LLM performance, as illustrated in Figure 1. Specifically, we identify core capability items linked to the overall task and its corresponding subtasks, then strengthen these capabilities through targeted knowledge injection and skill adaptation using a deliberately designed prompting strategy, Re-TASK prompting (see Figure 2). The capability items represent demonstrations of knowledge-skill adaptation, such as conceptual knowledge understanding and procedural knowledge applying. Notably, knowledge itself can be treated as a special capability item, as in the case of knowledge injection through knowledge recalling or retrieving. We then adopt in-context learning (ICL) techniques (Brown et al., 2020; Dong et al., 2022) to enhance the corresponding capabilities by carefully arranging these demonstrations within the prompt.

We conduct comprehensive experiments with open-source LLMs to evaluate the effectiveness of the Re-TASK framework in enhancing CoT performance across the law, finance, and mathematics domain tasks. By incorporating appropriate capability items that inject relevant domain knowledge or improve skill adaptation, we observe substantial improvements in task performance. Furthermore, we extend our experiments to include LLMs of varying scales, demonstrating that while model capabilities generally increase with scale, our framework can still effectively boost their performance.

---

[1]We use the terms "task" and "subtask" interchangeably, depending on the context.

Finally, we transition from manually identifying capability items to using automatically generated ones, further showcasing the broad applicability of the Re-TASK framework.

Our main contributions are summarized as follows:

- We introduce the Re-TASK framework, a novel theoretical model that revisits LLM tasks from the perspectives of capability, skill, and knowledge, offering a Chain-of-Learning view of tasks.

- Our research reveals that many failures of the CoT approach in addressing domain-specific tasks stem from insufficient knowledge or inadequate skill adaptation.

- We propose the Re-TASK prompting strategy, which integrates CoT with the Re-TASK framework to enhance LLM performance using in-context learning techniques.

- Extensive experimental results across the law, finance, and mathematics domains demonstrate the effectiveness of the Re-TASK framework. The automatic generation of capability items provides a scalable solution for enhancing LLM functionality in various domain-specific applications.

## 2  RELATED WORK

### 2.1  EDUCATIONAL THEORIES

Bloom's Taxonomy (Bloom, 2010) provides a foundational framework that links learning objectives with instructional activities. It posits that achieving learning goals requires the completion of multiple interconnected activities, categorized along two key dimensions: knowledge and cognitive processes. The knowledge dimension outlines four types of knowledge: factual, conceptual, procedural, and metacognitive. The cognitive process dimension establishes a hierarchy of cognitive skills, encompassing six levels—remember, understand, apply, analyze, evaluate, and create—each linked to specific cognitive processes, totaling 19 distinct actions. By integrating these dimensions, Bloom's Taxonomy serves as a comprehensive guide for educators to design curricula and instructional strategies that foster deeper understanding and encourage higher-order thinking in students.

Knowledge Space Theory (KST) (Doignon & Falmagne, 1985; Falmagne et al., 2013; Cosyn et al., 2021) offers a mathematical framework for modeling and assessing learners' knowledge within a specific domain. It identifies various knowledge states, defined as sets of problems or concepts that a learner can successfully solve or understand. The entirety of these possible knowledge states forms the knowledge structure, which delineates the relationships among different states. Learning pathways are the routes learners can take to transition from one knowledge state to another. By utilizing KST, educators can design effective educational interventions and personalized learning experiences, optimizing the learning process for each individual learner.

Building on these foundational educational theories, we propose the Re-TASK framework, which elucidates the dependence of tasks on various capability items. Each capability item is further broken down into its constituent aspects of knowledge and skills, highlighting the intricate relationships that contribute to task performance.

### 2.2  KNOWLEDGE AND SKILLS IN LLMs

Several studies have explored LLMs from the perspectives of knowledge and skill. KoLA (Yu et al., 2023) emphasizes the importance of world knowledge for LLMs and establishes a knowledge-oriented evaluation benchmark. In its approach to ability modeling, KoLA simplifies and selects from Bloom's learning theories to form four levels of knowledge-capability assessment: knowledge memorization, knowledge understanding, knowledge applying, and knowledge creating. Skill-it (Chen et al., 2024) posits that language models naturally acquire a sequence of skills from training data and formalizes the notion of a skill and an ordered set of skills in terms of associated data, differentiating this approach from traditional curriculum learning (Bengio et al., 2009), which focuses on training models using progressively difficult examples. In continual pre-training experiments, Skill-it's ordered learning of skills achieves faster convergence of validation loss compared to random sampling. RA-DIT (Lin et al., 2023) introduces a lightweight fine-tuning methodology that improves retrieval-augmented language models by enhancing both the relevance of retrieved knowledge and its effective utilization, marking a specialized form of knowledge and skill enhancement. MMLU (Hendrycks et al., 2020) serves as a benchmark designed to measure the possession of world

knowledge and problem-solving abilities. Reflecting on these developments, Bengio & Hu (2023) have emphasized the integration of the world model and the inference machine in current LLMs. They suggest that to reason effectively, a robust world knowledge model and a powerful inference machine are necessary, advocating for their separation and simultaneous development to enhance reasoning capabilities.

## 2.3 PROMPTING STRATEGIES

Various prompting strategies have been proposed to enhance model performance in solving complex, domain-specific tasks. One prominent approach is CoT (Wei et al., 2022; Kojima et al., 2022; Wang et al., 2023; Zhou et al., 2023), which decomposes a complex task into simpler subtasks, utilizing a divide-and-conquer strategy. Another notable method is Retrieval-Augmented Generation (RAG) (Lewis et al., 2020; Fan et al., 2024), which integrates neural language models with retrieval mechanisms to produce contextually relevant outputs by fetching knowledge from external sources. In-Context Learning (ICL) techniques (Brown et al., 2020; Dong et al., 2022) represent a significant advancement in prompting; ICL utilizes examples within the prompt itself, enabling the model to learn from context without the need for explicit retraining or fine-tuning.

In contrast, we propose Re-TASK prompting, which integrates CoT prompting with the Re-TASK framework. This approach leverages ICL techniques to enhance corresponding capabilities by carefully arranging demonstrations of capability items within the prompt, ultimately leading to improved overall task performance.

## 3 RE-TASK

### 3.1 RE-TASK FRAMEWORK

We begin by defining several key concepts within our framework: tasks, capability items, knowledge, and skills. We then elucidate how these elements interconnect to establish the structured Re-TASK framework, as illustrated in Figure 1(b). These concepts parallel Bloom's Taxonomy, where educational objectives—comparable to tasks in our framework—are systematically achieved through structured instructional activities, analogous to capability items. Each instructional activity involves the acquisition of specific types of knowledge and engages distinct cognitive processes, thereby facilitating knowledge-skill adaptation (or skill adaptation for short). The successful completion of an educational objective, or task, depends on mastering various capability items, each developed through these instructional activities. Knowledge Space Theory (KST) further reinforces this structured approach by highlighting sequential dependencies among learning items (akin to capability items in our framework) and establishing "learning pathways".

**Definition 1. (Task)** A task $\mathbf{T}$ is defined as a specific objective that LLMs are designed to achieve, characterized by a mapping from input $x$ to output $y$, facilitated by a task instruction $I$ and an optional context $ctx$. Formally, this relationship is expressed as $\mathbf{T}(ctx; I; x) = y$, where the semicolon denotes concatenation of inputs.

The optional context $ctx$ can be leveraged for knowledge injection or skill adaptation. By identifying corresponding capability items and incorporating a list of capability item demonstrations into $ctx$, this method aligns closely with in-context learning. In the Chain-of-Thought (CoT) paradigm, a task can be decomposed into a series of subtasks.

**Definition 2. (Knowledge)** A knowledge point $\mathbf{K}$ is defined as a text segment containing domain-specific knowledge that is essential for the performance of a task/subtask. In the context of LLMs, the knowledge $\mathbf{K}$ can also be implicit knowledge encoded within the model's parameters.

According to Bloom's Taxonomy, we consider three types of domain knowledge: factual, conceptual, and procedural[2]. Each type plays a distinct role in task execution, contributing differently to the LLM's ability to process and respond to task-specific demands.

**Definition 3. (Skill)** A skill $\mathbf{S}$ corresponds to the cognitive processes in Bloom's Taxonomy and is developed through related instructional activities, including knowledge recalling/retrieving, understanding, applying, and others.

---

[2]Metacognitive knowledge is beyond the scope of LLMs and is not our primary focus.

**Definition 4. (Capability Item)** A capability item $\mathbf{C}$, corresponding to the concept of instructional activities in Bloom's Taxonomy, is a specific exercise or demonstration designed to guide LLMs in applying a particular skill $\mathbf{S}$ to the relevant knowledge $\mathbf{K}$, thereby facilitating knowledge-skill adaptation.

Successfully completing a task $\mathbf{T}$ requires the sequential mastery of multiple capability items. These items can be conceptualized as a chain of learning, where dependencies among them are clearly defined. Figure 1(c) illustrates these dependencies. A task generally involves overall procedural knowledge ($C_{01}$), with its resolution corresponding to a capability item ($C_{02}$) that applies this knowledge in a manner akin to a CoT process. This procedural knowledge is further segmented into steps (i.e., subtasks), each linked to specific knowledge and involving different capability items $C_{ij}$.

Note that the knowledge $\mathbf{K}$ can be treated as a special capability item with the default skill of knowledge recalling or retrieving. Consequently, the task and its subtasks depend on two types of capability items: knowledge recalling (the knowledge itself) and knowledge-skill adaptation (e.g., knowledge understanding and applying).

## 3.2 Capability Item Construction

To effectively improve the performance of LLMs in domain-specific tasks, identifying key capability items is crucial. Our primary focus is on retrieving relevant knowledge (i.e., knowledge injection) and enhancing the understanding of conceptual knowledge along with the applying of procedural knowledge (i.e., knowledge-skill adaptation). These capability items are systematically designed to target specific aspects of task performance and are integral to the successful implementation of our Re-TASK framework. Below, we present several exemplary capability items that illustrate this strategic approach:

1) Knowledge retrieval: This involves identifying the relevant knowledge points for a given task or subtask and retrieving them from external sources. It may also include recalling internal knowledge points stored within the LLM.

2) Instances of conceptual knowledge: This involves understanding conceptual knowledge in real-world situations and providing an example that illustrates the conceptual knowledge. This example can help deepen understanding of the conceptual knowledge.

3) Execution of procedural knowledge: This capability is crucial for tasks that require following a set of ordered steps or procedures, such as technical troubleshooting, recipe preparation, or complex calculations. This capability item is an applying case of using procedural knowledge.

It is noteworthy that identifying the capability items associated with a task is a complex undertaking. For each given task, we can either manually pinpoint the relevant knowledge points and subsequently identify the corresponding skills that facilitate task resolution, or leverage LLMs to automate the entire process.

## 3.3 Re-TASK Prompting

By combining CoT with the Re-TASK framework, we can identify core capability items linked to the overall task and its corresponding subtasks, subsequently strengthening these capabilities to enhance subtask performance and, ultimately, overall task performance. Specifically, we enhance the capability items through targeted knowledge injection and skill adaptation using a deliberately designed prompting strategy known as Re-TASK prompting, as illustrated in Figure 2. The capability items serve as demonstrations of knowledge-skill adaptation, including knowledge recalling/retrieving, knowledge understanding, and knowledge applying.

We carefully arrange the demonstrations within the prompt according to their dependencies, following the chain of learning. Specifically, we sequence the capability items $C_{ij}$ for each subtask $i$. If multiple items are associated with the same subtask, we prioritize the knowledge itself (i.e., the capability item of knowledge retrieval) first, followed by more advanced items such as knowledge understanding and applying. Finally, we include the overall procedural knowledge $C_{01}$ along with its applying $C_{02}$ at the end of the prompt.

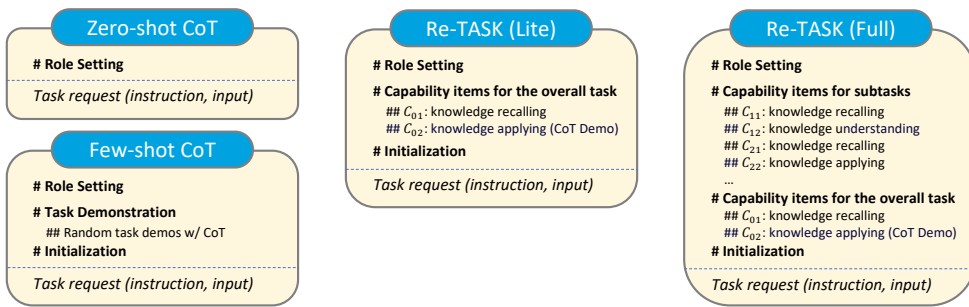

Figure 2: Comparison of prompting strategies: Zero-shot CoT, Few-shot CoT, Re-TASK (Lite) prompting with only capability items for the overall task, and Re-TASK (Full) prompting incorporating all available capability items. In Re-TASK prompting, a task or subtask may be associated with any number of capability items. Note that other strategies, such as Self-Consistency (SC), are excluded for brevity.

# 4 EXPERIMENTAL SETUP

To validate the effectiveness of Re-TASK, we constructed three datasets across the law, finance, STEM domains. We created a sentencing prediction dataset in the law domain, a financial course examination dataset in the finance domain and a mathematical reasoning dataset in the mathematics domain, leveraging larger LLMs to automatically generate capability items for the tasks.

It is worth noting that there are multiple approaches for constructing capability items, including manual design, retrieval-based methods (e.g., RAG), and direct generation using LLMs. We utilized LLMs to generate capability items because the primary focus of this paper is to validate the effectiveness of the Re-TASK framework, rather than to explore alternative methods for acquiring knowledge and capability items. This approach allows us to concentrate on evaluating the effectiveness of the Re-TASK framework using a consistent and comparable methodology.

## 4.1 SETTINGS

Given that both the legal and financial datasets are in Chinese, we opted for popular Chinese LLMs. Specifically, we selected the chat versions of Qwen1.5 (Bai et al., 2023), Llama3-Chinese (Cui, 2023), and Yi-1.5 (Young et al., 2024) for validation. For the mathematics dataset, which is in English, we employed a different set of LLMs, including Llama3 (Dubey et al., 2024), Mistral (Jiang et al., 2023), and Qwen1.5, to validate our results. To further verify the scalability of our framework, we conducted experiments with LLMs of varying scales, using the Qwen1.5 series with 7B, 14B, and 32B parameters on the legal dataset.

We developed Re-TASK prompting strategies that integrate demonstrations of capability items to validate their effectiveness in improving the performance of CoT on domain-specific tasks. Our baselines include Zero-shot CoT, Few-shot CoT, Plan-and-Solve and STEP-BACK. We evaluated the performance of Few-shot CoT under two settings: 1-shot and 3-shot learning, to align with the Re-TASK (Lite) and Re-TASK (Full) strategies, respectively. Additionally, we evaluated the performance of Zero-shot CoT with self-consistency. The prompt templates are illustrated in Figure 2.

## 4.2 TASKS AND DATASETS

We choose the sentencing prediction task in the law domain, the financial course examination task in the financial domain and the multiple choice question tasks in STEM for the validation of Re-TASK.

The sentencing prediction task in the law domain involves evaluating criminal offenders based on the factual descriptions provided in criminal cases and predicting the appropriate sentencing range for their judgments. This judgment process requires a high level of understanding and applying of relevant legal statutes and knowledge, demanding that LLMs possess a comprehensive grasp of legal principles, key case elements, essential sentencing concepts, and procedural logic in sentencing. As

Table 1: Comparison of accuracy (%) across Zero-shot CoT, Few-shot CoT, and Re-TASK strategies in the law domain. "Zero-shot CoT + SC" refers to Zero-shot CoT with self-consistency, and "n-shot CoT" refers to Few-shot CoT with $n$ randomly selected demonstrations.

|  |  | Llama3-Chinese-8B | Yi-1.5-9B | Qwen1.5-7B | Average Gain |
|---|---|---|---|---|---|
| Traditional CoT | Zero-shot CoT | 54.00 | 40.00 | 33.50 | - |
|  | Zero-shot CoT + SC | 54.50 | 40.50 | 33.50 | +0.33 |
|  | 1-shot CoT | 53.67 | 66.50 | 36.17 | +9.61 |
|  | 3-shot CoT | 56.33 | 70.17 | 38.50 | +12.50 |
| Baseline | Plan-and-Solve | 54.50 | 33.50 | 45.00 | +1.83 |
|  | Step-Back | 72.50 | 72.50 | 36.50 | +18.00 |
| **Re-TASK** | **Re-TASK (Lite)** | **78.50** | **85.00** | **45.50** | **+27.17** |

such, it serves as an ideal task for validating the Re-TASK framework. To construct the dataset, we utilized the publicly available CAIL dataset (China AI Law Challenge) (Xiao et al., 2018) and randomly sampled a test set comprising 200 instances.

Additionally, we selected the financial course examination task in the FinanceIQ dataset Zhang et al. (2023). The FinanceIQ dataset assesses LLMs' knowledge and reasoning abilities in financial contexts, evaluating their grasp of domain-specific knowledge. The financial dataset consists of multiple-choice questions, and the test set for the FinanceIQ dataset contains 178 instances.

We conducted experiments on the MMLU-Mathematics, Biology, and Physics benchmarks (Hendrycks et al., 2020). These datasets focus on evaluating mathematical reasoning, biological reasoning, and physical reasoning capabilities. Each dataset consists of multiple-choice questions. During the construction of the datasets, a few questions fell into an infinite cycle when generating capability items. As a result, we removed these instances and constructed test sets comprising 276, 144, and 102 instances for the mathematics, biology, and physics datasets, respectively.

### 4.3 CONSTRUCTION OF CAPABILITY ITEMS

We utilized large language models (LLMs) to assist in decomposing tasks and generating the capability items involved in Re-TASK. First, we predefined the capability types for each task based on the definition in Section 3.2 and then leveraged larger LLMs, which are presumed to possess sufficient domain knowledge, to automatically generate these items.

We began by employing LLMs to decompose the tasks, which resulted in the identification of the overall procedural knowledge ($C_{01}$). Next, we instructed the LLMs to create a CoT demonstration using the generated knowledge as a knowledge applying capability item ($C_{02}$). Task decomposition also guides the creation of capability items for each subtask. For each subtask $i$, we instructed the LLMs to generate relevant conceptual or procedural knowledge ($C_{i1}$). For conceptual knowledge, we requested illustrative examples to enhance understanding, while for procedural knowledge, we sought CoT demonstrations to illustrate effective applying ($C_{i2}$).

For financial tasks, we followed the complete procedure to generate capability items for both the overall task and its corresponding subtasks. In contrast, for other tasks, where the subtasks are relatively straightforward and do not require complex knowledge, we only generated capability items for the overall tasks.

## 5 EXPERIMENTAL RESULTS

We conducted experiments across three domains to validate the effectiveness of Re-TASK.

### 5.1 LAW DOMAIN

The results of the sentencing prediction task are presented in Table 1. Notably, Re-TASK (Lite) achieves the best performance across all settings, surpassing Zero-shot CoT by an average of 27.17%. Re-TASK (Lite) also outperforms the self-consistency version of Zero-shot CoT by a

Table 2: Comparison of token length acorss various prompting strategies in the law domain.

| | | Llama3-Chinese-8B | Yi-1.5-9B | Qwen1.5-7B |
|---|---|---|---|---|
| Traditional CoT | Zero-shot CoT | 526 | 510 | 431 |
| | 1-shot CoT | 1691 | 1185 | 1007 |
| | 3-shot CoT | 3104 | 2292 | 2176 |
| **Re-TASK** | **Re-TASK (Lite)** | **1291** | **1071** | **967** |

Table 3: Comparison of accuracy (%) across different prompt strategies on FinanceIQ dataset.

| | | Llama3-Chinese-8B | Yi-1.5-9B | Qwen1.5-7B | Average Gain |
|---|---|---|---|---|---|
| Traditional CoT | Zero-shot CoT | 36.52 | 53.93 | 43.82 | - |
| | Zero-shot CoT + SC | 34.27 | 61.80 | 46.63 | +2.81 |
| | 1-shot CoT | 34.69 | 64.33 | 46.07 | +3.60 |
| | 3-shot CoT | 34.27 | 63.82 | 46.07 | +3.30 |
| Baseline | Plan-and-Solve | 30.34 | 66.29 | 41.01 | +1.12 |
| | Step-Back | 30.90 | 66.85 | 44.38 | +2.62 |
| **Re-TASK** | **Re-TASK (Lite)** | **38.20** | **61.80** | **49.44** | **+5.06** |
| | **Re-TASK (Full)** | **52.81** | **73.60** | **51.69** | **+14.61** |

substantial margin of 26.84% on average. Specifically, Re-TASK (Lite) demonstrates a remarkable improvement of 45.00% with the Yi-1.5-9B model. Furthermore, when the knowledge item ($C_{01}$) is excluded from the demonstration count, Re-TASK (Lite), with a single demonstration, shows significant gains compared to the 1-shot CoT strategy. Step-Back also achieved a notable improvement, outperforming Zero-shot CoT by 18%, though it still lags behind Re-TASK (Lite) by 9.17%. These results underscore the effectiveness of the Lite version of Re-TASK prompting in the legal domain.

We further compared the token lengths generated by different prompt strategies, including both the prompt and the completion, to assess efficiency. As shown in Table 2, the inclusion of demonstrations generally resulted in an increase in token length. Specifically, the token length of Re-TASK (Lite) was comparable to that of 1-shot CoT and shorter than that of 3-shot CoT.

## 5.2 FINANCIAL DOMAIN

As shown in Table 3, Re-TASK exhibits substantial performance gains across all models in the FinanceIQ task. Notably, Re-TASK (Lite) and Re-TASK (Full) achieve average improvements of 5.06% and 14.61% over Zero-shot CoT, respectively. Moreover, Re-TASK (Lite) outperforms 1-shot CoT by 1.46%, and Re-TASK (Full) surpasses 3-shot CoT by 11.31%, highlighting the value of our added capability items. Furthermore, both Re-TASK (Lite) and Re-TASK (Full) significantly outperform the Plan-and-Solve and Step-Back methods.

Table 4: Comparison accuracy (%) across different prompt strategies on MMLU-Math dataset.

| | | Llama3-8B | Mistral-7B | Qwen1.5-7B | Average Gain |
|---|---|---|---|---|---|
| Traditional CoT | Zero-shot CoT | 40.58 | 24.28 | 36.96 | - |
| | Zero-shot CoT+SC | 48.19 | 24.64 | 41.67 | +4.23 |
| | 1-shot CoT | 49.42 | 23.41 | 36.52 | +2.51 |
| Baseline | Plan-and-Solve | 40.58 | 30.43 | 23.91 | -2.30 |
| | Step-Back | 45.65 | 34.42 | 19.93 | -0.60 |
| **Re-TASK** | **Re-TASK (Lite)** | **51.81** | **28.99** | **43.84** | **+7.61** |

Table 5: Comparison accuracy (%) across different prompt strategies on MMLU-Biology dataset.

|  |  | Llama3-8B | Mistral-7B | Qwen1.5-7B | Average Gain |
|---|---|---|---|---|---|
| Traditional CoT | Zero-shot CoT | 76.39 | 57.64 | 59.72 | - |
|  | Zero-shot CoT+SC | 78.47 | 60.42 | 62.50 | +2.55 |
|  | 1-shot CoT | 79.86 | 68.75 | 60.42 | +5.09 |
| Baseline | Plan-and-Solve | 73.61 | 55.56 | 61.11 | -1.16 |
|  | Step-Back | 43.75 | 50.69 | 53.47 | -15.28 |
| **Re-TASK** | **Re-TASK (Lite)** | **88.19** | **79.17** | **81.25** | **+18.29** |

Table 6: Comparison accuracy (%) across different prompt strategies on MMLU-Physics dataset.

|  |  | Llama3-8B | Mistral-7B | Qwen1.5-7B | Average Gain |
|---|---|---|---|---|---|
| Traditional CoT | Zero-shot CoT | 57.84 | 37.25 | 42.16 | - |
|  | Zero-shot CoT+SC | 58.82 | 39.22 | 37.25 | -0.65 |
|  | 1-shot CoT | 60.78 | 45.10 | 42.16 | +3.59 |
| Baseline | Plan-and-Solve | 55.88 | 42.16 | 41.18 | +0.65 |
|  | Step-Back | 30.39 | 34.31 | 30.39 | -14.05 |
| **Re-TASK** | **Re-TASK (Lite)** | **60.78** | **44.12** | **50.98** | **+6.21** |

### 5.3 STEM Domain

The results for MMLU-Math, MMLU-Biology and MMLU-Physics are presented in Table 4, Table 5 and Table 6. Re-TASK (Lite) demonstrates significant performance gains across all three datasets, particularly in the biology domain, where it outperforms Zero-shot CoT by an average of 18.29%. The performance of Step-Back is less favorable, likely due to the fact that the principles in STEM tasks are initially generated by the model itself. For smaller models, the quality of the generated principles is often suboptimal, leading to poorer final results. Re-TASK (Lite) surpasses 1-shot CoT in three datasets, highlighting its robust performance in STEM tasks.

Compared to the results in the legal domain, the improvements in the financial and STEM datasets are relatively modest. This may be attributed to the fact that the capability items automatically generated by LLMs, which, while effective, may not be fully optimized. Nonetheless, our methods show that leveraging knowledge and capability items generated from larger models can be used to augment the performance of smaller models. Since our primary objective is to demonstrate the potential of our approach rather than to optimize the identification of capability items, these results still underscore the value of our method.

## 6 Conclusion

In this paper, we introduced the Re-TASK framework, a novel approach that revisits LLM tasks through the lenses of capability, skill, and knowledge, aiming to address the limitations of the Chain-of-Thought (CoT) paradigm in complex, domain-specific tasks. By integrating Re-TASK with CoT, we systematically enhanced the performance of LLMs through targeted knowledge injection and skill adaptation using a structured prompting strategy, Re-TASK prompting. Our extensive experiments across the law, finance, and mathematics domains demonstrated that the Re-TASK framework significantly improves task performance, achieving substantial gains over baseline models and confirming the framework's potential to enhance LLM capabilities across diverse domains. Our research opens new avenues for practitioners to deepen their understanding, evaluation, and improvement of LLMs.

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

# A ALL EXPERIMENTAL RESULTS

## A.1 MODEL SCALING

Table 7: Accuracy comparison (%) across different prompt strategies using various scales of Qwen1.5 models.

|  | Qwen1.5-7B | Qwen1.5-14B | Qwen1.5-32B |
|---|---|---|---|
| Zero-shot CoT | 33.50 | 48.50 | 84.00 |
| **Re-TASK (Lite)** | **36.83** | **74.67** | **84.33** |
| **Re-TASK (Full)** | **56.00** | **80.33** | **89.50** |

The accuracy results for different prompt strategies across various scales of Qwen1.5 models on the sentencing prediction dataset are presented in Table 7.

## A.2 EFFICIENCY

Table 8: Comparison of token length across various prompting strategies on FinanceIQ dataset

|  | Llama3-Chinese-8B | Yi-9B | Qwen1.5-7B |
|---|---|---|---|
| Zero-shot CoT | 861 | 541 | 417 |
| 1-shot CoT | 1138 | 844 | 767 |
| 3-shot CoT | 1730 | 1478 | 1421 |
| **Re-TASK (Lite)** | **1032** | **742** | **602** |
| **Re-TASK (Full)** | **1713** | **1707** | **1581** |

Table 9: Comparison of token length across various prompting strategies on MMLU-Math dataset.

|  | Llama3-8B | Qwen1.5-7B | Mistral-7B |
|---|---|---|---|
| Zero-shot CoT | 301 | 449 | 544 |
| 1-shot CoT | 615 | 615 | 768 |
| **Re-TASK(Lite)** | **628** | **673** | **769** |

In order to explore the efficiency of Re-TASK and avoid the impact of computing platform performance on efficiency evaluation, we use the sum of tokens for input and output of the LLMs as the metric of efficiency. Table 8 and Table 9 illustrate the average length of tokens with different methods on FinanceIQ and MMLU-Math dataset across all questions. It can be seen that the number of tokens for Re-TASK and the corresponding 1-shot or 3-shot CoT are similar. Therefore the efficiency is also comparable. While the accuracy of Re-TASK is significantly higher, it does not reduce efficiency substantially.

## A.3 CASE STUDY

We also conduct the case study on FinanceIQ and MMLU-Math datasets.

Figures 3, 4, and 5 show the results of the same input-output example using three prompt strategies—Zero-shot CoT, Re-TASK (Lite), and Re-TASK (Full)—on the FinanceIQ dataset with the LLaMA3-8B-Chinese model. From the case, we can see that in the Zero-shot CoT approach, the model fails to produce the correct answer because it lacks knowledge of the problem-solving steps and relevant knowledge about the topic, leading to repeated attempts and ultimately an incorrect result. In Re-TASK (Lite), while the model knows the correct formula for solving the problem, it makes errors during the numerical calculations, resulting in an incorrect answer as well. In Re-TASK (Full), the model applies the correct formula and arrives at the correct answer. This indicates

---

**Prompt and Response for Standard Zero-shot CoT on FinanceIQ Dataset.**

---

**# Role:** 你是一名经济领域的专家，请完成指定的任务。
(**# Role:** You are an expert in the field of economics, please complete the tasks specified by the user.)

**任务描述**：现在根据下列输入的问题，从4个选项中选择一个正确的选项，请将答案填在[选项]与[选项]之间，例如[选项]C[选项]，必须给出一个选项。
(**Task description:** Now, based on the following input question, select the correct option from the 4 options. Please fill in the answer between [options] and [options], for example [option] C [option], one option must be given.)

**问题 (Question):**
以5亿作为基础货币，当活期存款准备金率是10%，定期存款准备金率是2%，定期存款比率达到30%，超额准备金率为5%，通货比率为20%时，你知道货币存量（以亿为单位）应该是多少吗？
(With 500 million as the base currency, when the demand deposit reserve ratio is 10%, the fixed deposit reserve ratio is 2%,the fixed deposit ratio is 30%, the excess reserve ratio is 5%, and the currency ratio is 20%, Do you know what the currency stock (in billion) should be?)
**选项 (Options):** A:16.20, B:16.85, C:12.45, D:9.23

**答案 (Answer):**
首先，我们需要计算货币存量的基础值。货币存量的基础值=基础货币/(1+活期存款准备金率+定期存款准备金率+定期存款比率+超额准备金率+通货比率)。货币存量的基础值=5亿/(1+0.1+0.02+0.3+0.05+0.2)=5亿/1.67=3亿。因此，货币存量的基础值为3亿。

接下来，我们需要计算货币存量的超额准备金率。货币存量的超额准备金率=超额准备金率/(1+活期存款准备金率+定期存款准备金率+定期存款比率+通货比率)。货币存量的超额准备金率=0.05/(1+0.1+0.02+0.3+0.2)=0.05/1.67=3%。因此，货币存量的超额准备金率为3%。最后，我们需要计算货币存量的货币存量。货币存量=基础货币+超额准备金率*货币存量的超额准备金率。货币存量=5亿+0.05*3亿=5亿+1500万=5.15亿因此，货币存量为5.15亿。

(First, we need to calculate the base value of the money stock. Base value of the currency stock = Base currency/ (1 + demand deposit reserve ratio + fixed deposit reserve ratio + fixed deposit reserve ratio + excess reserve ratio + currency ratio). The base value of the money stock = /(1+0.1+0.02+0.3+0.05+0.2)=5 million / 167 million = 300 million. Therefore, the base value of the money stock is 300 million

Next, we need to calculate the excess reserve ratio of the money stock. Excess Reserve Ratio of Money Stock = Excess ReserveRatio/ (1 + Demand Deposit Reserve Ratio + Fixed Deposit Reserve Ratio + Fixed Deposit Reserve Ratio + Currency Ratio). The excess reserve ratio of the money stock is =0.05/(1+0.1+0.02+0.3+0.2)=0.05/1.67=3% . As a result, the excess reserve ratio for the money stock is 3%.

Finally, we need to calculate the money stock of the money stock. Currency stock = Base currency + Excess reserve ratio * Excess reserve ratio of currency stock. Money stock = 500 million + 0.05 * 300 million = 500 million + 15 million = 515 million. Therefore, the money stock is 515 million.)

---

Figure 3: An example of the Prompt and the Response for Zero-shot CoT on FinanceIQ Dataset.

that adding different capability items supplements the model's knowledge and enhances its abilities in domain tasks, demonstrating the effectiveness of the Re-TASK framework.

Table 10 and 11 show one pair of comparison of Zero-shot CoT and Re-TASK (Lite) with Llama3-8B on MMLU-Math dataset. For the Zero-shot CoT, it can be seen that though the LLM can stimulate the knowledge of the Pythagorean theorem, the following reasoning and calculation is still wrong. For the Re-TASK, we incorporate knowledge of the Pythagorean theorem through the prompt and demonstrated the applying of this knowledge through one capability item, and the correct reasoning and calculation result can be obtained. This comparison intuitively represents that capability items can effectively improve the reasoning.

# B EXPERIMENTAL MODELS AND DATASETS

## B.1 MODELS

Five models are employed in this work: Qwen1.5-7B-Chat, Qwen1.5-14B-Chat, Qwen1.5-32B-Chat, Yi-1.5-9B-Chat, and LLaMA-Chinese-8B-Instruct. The Qwen1.5 series and Yi-1.5 series models are obtained as official chat versions from Hugging Face, while the LLaMA3-Chinese-8B model is downloaded from Modelscope, as shown in Table 12.

The sources of the datasets used in this paper are shown in Table 13.

## B.2 SENTENCING PREDICTION DATASETS

We utilize the dataset from the Cail2018 competition(Xiao et al., 2018), sourced from publicly available criminal legal documents on "China Judgments Online". Each record in the dataset comprises descriptions of case details and factual parts extracted from the legal documents. Furthermore, each case includes applicable legal articles, the charges against the defendant, and the length of the sentence. The dataset encompasses approximately 2.68 million criminal law documents, involving 183

---

**Prompt and Response for Re-Task (Lite) on FinanceIQ Dataset.**

**# Role:** 你是一名金融领域的专家，请通过对<Knowledge Application>的学习加深对响应的<Knowledge Recall>的应用能力，完成用户指定的任务。
(**# Role:** You are an expert in the field of economics, please deepen your application ability of<Knowledge Recall>through learning<Knowledge Application> and complete the tasks specified by the user.)

**# Capability Items for Overall Task:**
**## Knowledge Recall**
为了解决这道题，需要先理解题意，然后根据题意计算货币乘数，最后计算货币存量，可以得到答案。
(To solve this problem, it is necessary to first understand the meaning of the question, then calculate the money multiplierbased on the meaning of the question, and finally calculate the money stock to obtain the answer.)

**## Knowledge Application**
**问题 (Question):**
假设基础货币为4亿，活期存款准备金率为8%，定期存款准备金率为3%，定期存款比率为25%，超额准备金率为4%，通货比率为15%。根据这些条件，计算货币存量（以亿为单位）应该是多少？
(Assuming the base currency is 400 million, the reserve requirement ratio for current deposits is 8%, the reserve requirementratio for fixed deposits is 3%, the fixed deposit ratio is 25%, the excess reserve requirement ratio is 4%, and the currency ratio is 15%. Based on these conditions, what should the currency stock (in billions) be calculated?)
**选项 (Options):** A:10.24, B:11.56, C:12.32, D:13.48
**答案 (Answer):** 首先，理解题意，题目要求计算在给定的基础货币、各种准备金率和通货比率条件下的货币存量。接着，计算货币乘数，货币乘数= 1 / (活期存款准备金率 + 定期存款准备金率 * 定期存款比率 + 超额准备金率 + 通货比率) = 1 / (0.08 + 0.03 * 0.25 + 0.04 + 0.15) = 1 / (0.08 + 0.0075 + 0.04 + 0.15) = 1 / 0.2775 ≈ 3.604。最后，计算货币存量，货币存量 = 基础货币 * 货币乘数 = 4 * 3.604 = 14.416。由于选项中没有14.416这个数值，最接近的是13.48，因此答案是[选项]D[选项]。但根据计算，最准确的答案应该是14.416，这表明选项中可能存在误差或需要重新评估选项设置。然而，基于提供的选项，最接近的答案是D。
(Firstly, understand the meaning of the question, which requires calculating the money stock under given conditions of base currency, various reserve ratios, and currency ratios. Next, calculate the currency multiplier, which is 1/(current deposit reserve ratio+fixed deposit reserve ratio * fixed deposit ratio+excess reserve ratio+currency ratio)=1/(0.08+0.03 * 0.25+0.04+0.15)=1/(0.08+0.0075+0.04+0.15)=1/0.2775 ≈ 3.604. Finally, calculate the moneystock, where money stock=base currency * money multiplier=4 * 3.604 ≈ 14.416. Since there is no value of 14.416 in the options, the closest value is 13.48so the answer is [option] D [option]. But according to calculations, the most accurate answer should be 14.416, indicating that there may be errors in the options or that the option settings need to be reevaluated. However, based on the provided options, the closest answer is D.)

**# Initialization**
作为<Role>，你可以参考<Demonstration>完成用户给定的任务。
(As a<Role>, you can refer to<Demonstration>to complete the tasks given by the user.)

**任务描述：** 现在根据下列输入的问题，从4个选项中选择一个正确的选项，请将答案填在[选项]与[选项]之间，例如[选项]C[选项]，必须给出一个选项。
(**Task description:** Now, based on the following input question, select the correct option from the 4 options. Please fill in the answer between [options] and [options], for example [option] C [option], one option must be given.)

**问题 (Question):**
以5亿作为基础货币，当活期存款准备金率是10%，定期存款准备金率是2%，定期存款比率达到30%，超额准备金率为5%，通货比率为20%时，你知道货币存量（以亿为单位）应该是多少吗？
(With 500 million as the base currency, when the demand deposit reserve ratio is 10%, the fixed deposit reserve ratio is 2%,the fixed deposit ratio is 30%, the excess reserve ratio is 5%, and the currency ratio is 20%, Do you know what the currency stock (in billions) should be?)
**选项 (Options):** A:16.20, B:16.85, C:12.45, D:9.23

**答案 (Answer):**
首先，我们需要计算货币存量的计算公式为：货币存量=基础货币/(1-活期存款准备金率-定期存款准备金率-超额准备金率)根据题目中的条件，我们可以计算货币存量：货币存量=5亿/(1-10%-2%-5%)=5亿/0.77=6.49亿所以，货币存量应该是6.49亿。答案：[选项]C[选项]
( First, we need to calculate the currency stock as follows: Currency stock = Base currency/ (1- Demand deposit reserve ratio - Time deposit reserve ratio - Excess reserve ratio) According to the conditions in the topic, we can calculate the money stock: Money stock = 500 million / (1- 10% - 2% - 5%) = 500 million / 0.77 = 649 million So, money stock should be 649 million. Therefore, the answer is C. )

---

Figure 4: An example of the Prompt and the Response for Re-TASK (Lite) CoT on FinanceIQ Dataset.

different charges and 202 legal articles, with sentences ranging from 0 to 25 years, life imprisonment, and the death penalty. Our focus is exclusively on the task of sentence prediction.

The sentence prediction task entails estimating the length of a defendant's sentence using the descriptions and facts provided in the criminal legal documents. This task integrates five essential elements from the dataset: the facts of the crime, charges, referenced legal articles, defendant's name, and the length of the sentence.

For our use of the CAIL2018 dataset, we implement the following processing steps:

1) Several commonly encountered criminal law articles are selected to serve as the knowledge base. Then, the original dataset is filtered to include only criminal cases relevant to these articles. Additionally, we exclude data instances that inherently implied sentence prediction months based on the range of the sentence.

2) We designed a suitable task instruction and standardized the output format of the task.

3) The specific month lengths of sentences are converted into three broader sentencing categories, ABC (A: under 3 years; B: 3 to 10 years; C: over 10 years).

4) A series of robust and effective test prompt templates are designed for the task.

From the CAIL 2018 data, four datasets are generated: a 200-instance training set, a 600-instance training set, a 200-instance test set, and a 200-instance capability item set. The 600-instance training set, 200-instance test set, and the 200-instance capability item set are independent, with no overlap-

**Prompt and Response for Re-Task (Full) on FinanceIQ Dataset.**

**# Role:** 你是一名金融领域的专家，请通过<Knowledge Example 1>的学习加深对响应的<Knowledge Recall 1>的理解能力，对通过对<Knowledge Application 2>的学习加深对响应的<Knowledge Recall 2>的应用能力，完成用户指定的任务。
(# Role: You are an expert in the field of economics, please deepen your understanding of <Knowledge Recall 1> through learning <Knowledge Example 1> and deepen your application ability of <Knowledge Recall> through learning <Knowledge Application> and complete the tasks specified by the user.)

**# Capability Items for Subtasks:**
**## Knowledge Recall 1**
基础货币、活期存款准备金率、定期存款准备金率、定期存款比率、超额准备金率、通货比率的概念及其对货币存量的影响。
(The concepts of base currency, reserve requirement ratio for current deposits, reserve requirement ratio for fixed deposits, fixed deposit ratio, excess reserve ratio, and currency ratio and their impact on the stock of money.)
**## Knowledge Example 1**
[Please Put the Knowledge Example 1 of Subtasks Here.]

**## Knowledge Recall 2**
货币存量 = 基础货币 / (活期存款准备金率 + 定期存款准备金率 * 定期存款比率 + 超额准备金率 + 通货比率)。
(Currency stock=base currency/(current reserve requirement ratio+fixed reserve requirement ratio * fixed deposit ratio+excess reserve requirement ratio+currency ratio).)
**## Knowledge Application 2**
[Please Put the Knowledge Application 2 of the Subtasks Here.]

**# Capability Items for Overall Task:**
**## Knowledge Recall**
为了解决这道题，需要先理解题意，然后根据题意计算货币乘数，最后计算货币存量，可以得到答案。
(To solve this problem, it is necessary to first understand the meaning of the question, then calculate the money multiplier based on the meaning of the question, and finally calculate the money stock to obtain the answer.)
**## Knowledge Application**
**问题 (Question):**
假设基础货币为4亿，活期存款准备金率为8%，定期存款准备金率为3%，定期存款比率为25%，超额准备金率为4%，通货比率为15%。根据这些条件，计算货币存量（以亿为单位）应该为多少？
(Assuming the base currency is 400 million, the reserve requirement ratio for current deposits is 8%, the reserve requirement ratio for fixed deposits is 3%, the fixed deposit ratio is 25%, the excess reserve requirement ratio is 4%, and the currency ratio is 15%. Based on these conditions, what should the currency stock (in billions) be calculated?)
**选项 (Options):** A:10.24, B:11.56, C:12.32, D:13.48
**答案 (Answer):** 首先，理解题意，题目要求计算在给定的基础货币、各种准备金率和通货比率条件下的货币存量。接着，计算货币乘数，货币乘数= 1 / (活期存款准备金率 + 定期存款准备金率 * 定期存款比率 + 超额准备金率 + 通货比率) = 1 / (0.08 + 0.03 * 0.25 + 0.04 + 0.15) = 1 / (0.08 + 0.0075 + 0.04 + 0.15) = 1 / 0.2775 ≈ 3.604 最后，计算货币存量，货币存量 = 基础货币 * 货币乘数 = 4 * 3.604 ≈ 14.416。由于选项中没有14.416这个数值，最接近的是13.48，因此答案是[选项]D[选项]。但根据计算，最准确的答案应该是14.416，这表明选项中可能存在误差或需要重新评估选项设置。然后，基于提供的选项，最接近的答案是D。
(Firstly, understand the meaning of the question, which requires calculating the money stock under given conditions of base currency, various reserve ratios, and currency ratios. Next, calculate the currency multiplier, which is 1/(current deposit reserve ratio+fixed deposit reserve ratio * fixed deposit ratio+excess reserve requirement ratio+currency ratio)=1/(0.08+0.03 * 0.25+0.04+0.15)=1/(0.08+0.0075+0.04+0.15)=1/0.2775 ≈ 3.604. Finally, calculate the money stock, where money stock=base currency * money multiplier=4 * 3.604 ≈ 14.416. Since there is no value of 14.416 in the options, the closest value is 13.48, so the answer is [option] D [option]. But according to calculations, the most accurate answer should be 14.416, indicating that there may be errors in the options or that the option settings need to be re evaluated. However, based on the provided options, the closest answer is D.)

**# Initialization**
作为<Role>，你可以参考<Demonstration>完成用户给定的任务。
(As a <Role>, you can refer to <Demonstration> to complete the tasks given by the user.)

**任务描述：**现在根据下列输入的问题，从4个选项中选择一个正确的选项，请将答案填在[选项]与[选项]之间，例如[选项]C[选项]，必须给出一个选项。
(Task description: Now, based on the following input question, select the correct option from the 4 options. Please fill in the answer between [options] and [options], for example [option] C [option], one option must be given.)

**问题 (Question):**
以5亿作为基础货币，当活期存款准备金率是10%，定期存款准备金率是2%，定期存款比率达到30%，超额准备金率为5%，通货比率为20%时，你知道货币存量（以亿为单位）应该是多少吗？
(With 500 million as the base currency, when the demand deposit reserve ratio is 10%, the fixed deposit reserve ratio is 2%, the fixed deposit ratio is 30%, the excess reserve ratio is 5%, and the currency ratio is 20%, Do you know what the currency stock (in billions) should be?)
**选项 (Options):** A:16.20, B:16.85, C:12.45, D:9.23

**答案 (Answer):**
首先根据货币乘数公式：货币乘数=1/(活期存款准备金率+定期存款准备金率*定期存款比率+超额准备金率+通货比率)。将题目中的数值代入公式计算：
1/(0.10+0.02*0.30+0.05+0.20)=1/(0.10+0.006+0.05+0.20)=1/0.286=3.49 然后根据货币存量的计算公式：货币存量=基础货币*货币乘数。将题目中的数值代入公式得到答案是16.85亿元。因此，答案选B。
(First, according to the currency multiplier formula: currency multiplier = 1 / (Demand deposit reserve ratio + Fixed deposit reserve ratio * Fixed deposit ratio + Excess reserve ratio + Currency ratio). Substitute the numerical value in the question into the formula: 1/(0.10+0.02*0.30+0.05+0.20)=1/(0.10+0.006+0.05+0.20)=1/0.286=3.49 . Then, according to the calculation formula of currency stock: currency stock = base currency * currency multiplier. Substitute the value in the question into the formula and the answer is 1.685 billion yuan. Therefore, the answer is B.)

Figure 5: An example of the Prompt and the Response for Re-TASK (Full) CoT on FinanceIQ Dataset.

ping instances. The 200-instance training set is randomly sampled from the 600-instance training set. Additionally, the 200-instance capability item set is specifically created to develop capability items 2, 3, and 4. The distribution of the three sentencing options (A,B,C) in the four datasets approximates a 1:1:1 ratio.

Examples of sentence prediction task is illustrated in Figure6.

## B.3 FINANCEIQ AND MMLU-MATH DATASETS

For the MMLU-Math dataset, we utilized the original versions of the 304 questions, and we used the Llama3.1-70B to generate knowledge, and based on the effectiveness of the knowledge, we filtered out 276 questions from it. In contrast, we conducted some processing for the FinanceIQ dataset. Our analysis revealed that a substantial portion of the FinanceIQ dataset consists of knowledge-intensive questions that can be addressed solely by providing factual knowledge, without necessitating high levels of comprehension and reasoning abilities. Consequently, we initially employed the Qwen2.5-72B model to assist in a filtering process, which effectively eliminated simpler questions that could be answered directly through factual knowledge.

During the generation process, we encountered two special situations: 1. Some questions did not have explicit knowledge points and could be answered with simple deductions; 2. In some questions, the knowledge points were contained within the options rather than in the stem, resulting in the model generating knowledge points only for the correct answer and avoiding other incorrect answers. This can affect the subsequent validation of the Re-TASK framework. Due to these two situations, we removed this portion of the data.

We utilize Qwen2.5-72B to assist in the task decomposition of instances in the FinanceIQ dataset, while LLAMA3.1-70B is employed for the decomposition of tasks in the MMLU dataset. In our experiments, we found that the subtasks in the MMLU dataset were overly simplistic, resulting in only task-level knowledge and capability items. In contrast, the subtasks in FinanceIQ still present a certain level of difficulty, necessitating subtask-level knowledge and capability items. We continue

Table 10: An example of standard zero-shot CoT on MMLU-Math dataset.

| **Prompt and Response for Standard Zero-shot CoT** |
| --- |
| **Input:**
# Role:
You are an expert in the field of Math. Complete the task provided by the user.

# Task Description:
According to the following question and options, select one correct option. Please think step by step and follow the answer format, i.e., output the brief chain-of-thought beginning with 'Rationale: ' within 500 characters, and the final choice option beginning with 'Correct: ' from 'A' to 'D'.
Question:
The length of a rectangle is twice its width. Given the length of the diagonal is $5\sqrt5$, find the area of the rectangle.
Options:
A. 2500; B. 2; C. 50; D. 25

**Output:**
Rationale: Let the width of the rectangle be x. Since the length is twice the width, the length is 2x. The diagonal is the hypotenuse of a right triangle with legs x and 2x. Using the Pythagorean theorem, we can write:
x^2 + (2x)^2 = (5\sqrt5)^2
Simplifying the equation, we get:
5x^2 = 25
Dividing both sides by 5, we get:
x^2 = 5
Taking the square root of both sides, we get:
x = $\sqrt{5}$
Now, the area of the rectangle is:
Area = length × width = 2x × x = 2($\sqrt{5}$)^2 = 2(5) = 10
Correct: D. 25 |

to use Qwen2.5-72B to help generate these capability items. Detailed information on generating knowledge points and capability items can be found in the appendix.

### B.4 CAPABILITY ITEM DATASETS FOR SENTENCING PREDICTION TASK

We manually identified three capability items for the overall task ($C_{0x}$) and three capability items ($C_{1x}$ and $C_{2x}$) for subtasks in law domain:

$C_{01}$: This capability item pertains to knowledge recalling within the overall task, specifically covering the legal articles. In this experiment, $C_{01}$ includes all the knowledge needed for both subtask 1 and subtask 2, so we also used its content for $C_{11}$ and $C_{21}$. $C_{02}$: This capability item aims to enhance the LLM's understanding of legal articles through structured expression. $C_{03}$: This capability item pertains to the applying of legal articles, specifically demonstrating the CoT (Chain of Thought) version of the sentence prediction task. $C_{12}$: This capability item focuses on identifying key elements in the factual descriptions presented in criminal cases to enhance the LLM's understanding in subtask 1. $C_{13}$: This capability item involves the illustrative differentiation of essential sentencing concepts to enhance the LLM's applying of injury assessment knowledge in subtask 1. $C_{22}$: This capability item provides illustrative explanation of sentencing outcomes to enhance the LLM's ability to apply procedural knowledge in subtask 2.

The data for Capability item 1 originated from criminal law, selecting 100 frequently used legal articles as the initial data. The data for Capability items 2, 3, and 4 came from the 200-instance capability item set, with 100 instances randomly chosen as the original data.

The task templates created for the capability items are illustrated as Figure7, Figure 8, Figure 9 and Figure 10.

Table 11: An example of Re-TASK prompt on MMLU-Math dataset.

| **Prompt and Response for Re-TASK Prompting** |
|---|

**Input:**
# Role:
You are an expert in the field of Math. Complete the task provided by the user.

# Knowledge:
The area of a rectangle is given by A = length * width, and the Pythagorean theorem states that for a right triangle, a^2 + b^2 = c^2, where c is the length of the hypotenuse (diagonal).

# Demonstration:
Question: In a rectangle, the length is 5 cm and the width is 12 cm. What is the length of the diagonal?
Options:
A. 13 cm; B. 14 cm; C. 15 cm; D. 16 cm
Rationale:
Step 1. We're given a rectangle with length 5 cm and width 12 cm. Step 2. We can use the Pythagorean theorem to find the length of the diagonal. Step 3. The diagonal is the hypotenuse of a right triangle with legs 5 cm and 12 cm. Step 4. Using the theorem, we get diagonal^2 = 5^2 + 12^2 = 25 + 144 = 169. Step 5. Taking the square root, we get diagonal = 13 cm. The correct answer is A.
Correct: A

# Task Description:
According to the following question and options, refer to <# Knowledge> and <# Demonstration>, select one correct option. Please think step by step and follow the answer format in <# Demonstration>, i.e., output the brief chain-of-thought beginning with 'Rationale: ' within 500 characters, and the final choice option beginning with 'Correct: ' from 'A' to 'D'.
Question:
The length of a rectangle is twice its width. Given the length of the diagonal is $5\sqrt5$, find the area of the rectangle.
Options:
A. 2500; B. 2; C. 50; D. 25

**Output:**
Rationale:

Step 1. We're given a rectangle with length = 2 * width. Step 2. We're also given the length of the diagonal, which is $5\sqrt5$. Step 3. We can use the Pythagorean theorem to find the length and width. Step 4. Let the width be x, then the length is 2x. Step 5. Using the theorem, we get (2x)^2 + x^2 = ($5\sqrt5$)^2. Step 6. Simplifying, we get 4x^2 + x^2 = 125. Step 7. Combining like terms, we get 5x^2 = 125. Step 8. Dividing both sides by 5, we get x^2 = 25. Step 9. Taking the square root, we get x = 5. Step 10. Now we can find the area, which is length * width = 2x * x = 2 * 5 * 5 = 50.
Correct: C

## B.5 EXPERIMENTAL SETTINGS

# C EXPERIMENTAL PROMPTS

## C.1 LAW DOMAIN

The prompt templates using in the law domain are shown in Figure 11, Figure 12, Figure 13, Figure 14 and Figure 15.

Table 12: Models, Sources and Licenses Used in This Work

| Models | Model sources | License |
|---|---|---|
| Qwen1.5-7B | https://huggingface.co/Qwen/Qwen1.5-7B-Chat | Apache License 2.0 |
| Qwen1.5-14B | https://huggingface.co/Qwen/Qwen1.5-14B-Chat | Apache License 2.0 |
| Qwen1.5-32B | https://huggingface.co/Qwen/Qwen1.5-32B-Chat | Apache License 2.0 |
| Yi-1.5-9B | https://huggingface.co/01-ai/Yi-1.5-9B-Chat | Apache License 2.0 |
| Llama3-Chinese-8B | https://www.modelscope.cn/models/FlagAlpha/ Llama3-Chinese-8B-Instruct/summary | Apache License 2.0 |
| Llama3-8B | https://www.modelscope.cn/models/FlagAlpha/ Llama3-8B-Instruct/summary | Apache License 2.0 |
| Mistral-7B | https://huggingface.co/mistralai/Mistral-7B-Instruct-v0.2 | Apache License 2.0 |

Table 13: Datasets, sources and licenses used in this work

| Datasets | Sources |
|---|---|
| MMLU | https://huggingface.co/datasets/cais/mmlu |
| FinanceIQ | https://huggingface.co/datasets/Duxiaoman-DI/FinanceIQ |
| CAIL 2018 | https://github.com/thunlp/CAIL?tab=readme-ov-file |

## C.2 MATH DOMAIN

The prompt templates using in the math domain are shown in Table 14, Table 15, Table 16, Table 17 and Table 18.

Table 14: The prompt template of standard zero-shot CoT on MMLU-Math dataset.

**Prompt template of Standard Zero-shot CoT**

# Role:
You are an expert in the field of Math. Complete the task provided by the user.

# Task Description:
According to the following question and options, select one correct option. Please think step by step and follow the answer format, i.e., output the brief chain-of-thought beginning with 'Rationale: ' within 500 characters, and the final choice option beginning with 'Correct: ' from 'A' to 'D'.
Question:
[Please Put Your Question Here]
Options:
[Please Put Your Options Here]

## C.3 FINANCIAL DOMAIN

The prompt templates using in the financial domain are shown in Figure 16, Figure 17, Figure 18 and Figure 19.

**Examples**

| | |
|---|---|
| **Instruction1** | 任务描述：根据下列输入，对被告人陈某进行刑期判决。
(Task description: According to the following input, sentence the defendant Chen.)
要求：从A,B,C三个刑期区间选项(A:三年以下有期徒刑、拘役或者管制；B:三年以上十年以下有期徒刑；C:十年以上有期徒刑、无期徒刑或死刑)中选出一个最合适的判决刑期区间，请将答案填在刑期区间与[eoa]之间，必须给出一个选项，请严格按要求的输出格式输出答案，不要输出任何无关内容或者解释。例如刑期区间]C[eoa]。
(Requirement: Select the most appropriate sentence interval from the three sentence interval options A, B, and C (A: imprisonment of less than three years, detention or control; B: imprisonment of more than three years but less than ten years; C: imprisonment of more than ten years, life imprisonment or death penalty). Please fill in the answer between [SENTENCE] and [eoa]. You must give an option. Please output the answer strictly in the required output format. Do not output any irrelevant content or explanation. For example, [SENTENCE] C [eoa].) |
| **Input1** | 淄博市张店区人民检察院指控，2013年10月21日8时许，被告人张某、张2某、陈某受栾某（已判刑）指使，对刘某实施殴打致刘某伤情构成轻伤二级。
(The People's Procuratorate of Zhangdian District, Zibo City, accused that at about 8:00 on October 21, 2013, the defendants Zhang, Zhang 2, and Chen were instructed by Luan (who has been sentenced) to beat Liu, causing Liu's injury to constitute a second-degree minor injury. The defendant Chen's behavior constitutes ××. The public prosecution agency has transferred relevant evidence to this court and requested this court to investigate the criminal responsibility of the defendant Chen in accordance with the provisions of ×××× of the Criminal Law of the People's Republic of China.) |
| **Output1** | [刑期区间]A[eoa] ([SENTENCE] A [eoa]) |
| **Instruction2** | 任务描述：根据下列输入，对被告人唐某进行刑期判决。
(Task description: According to the following input, sentence the defendant Tang.)
要求：从A,B,C三个刑期区间选项(A:三年以下有期徒刑、拘役或者管制；B:三年以上十年以下有期徒刑；C:十年以上有期徒刑、无期徒刑或死刑)中选出一个最合适的判决刑期区间，请将答案填在刑期区间与[eoa]之间，必须给出一个选项，请严格按要求的输出格式输出答案，不要输出任何无关内容或者解释。例如刑期区间]C[eoa]。
(Requirement: Select the most appropriate sentence interval from the three sentence interval options A, B, and C (A: imprisonment of less than three years, detention or control; B: imprisonment of more than three years but less than ten years; C: imprisonment of more than ten years, life imprisonment or death penalty). Please fill in the answer between [SENTENCE] and [eoa]. You must give an option. Please output the answer strictly in the required output format. Do not output any irrelevant content or explanation. For example, [SENTENCE] C [eoa].) |
| **Input2** | 2016年9月25日7时许，被告人唐某与黄某因公路堡坎问题发生口角，继而互相抓打。唐某手持锄头打向黄某，锄头打中正在劝架的被害人谭某后脑，致谭某当场昏迷。唐某拨打"110"电话报警，到案后如实供述了事发经过。经鉴定，被害人谭某损伤程度系重伤二级。
(At about 7:00 a.m. on September 25, 2016, the defendant Tang and Huang had an argument over a road embankment, and then they fought each other. Tang hit Huang with a hoe, and the hoe hit the back of the head of the victim Tan who was trying to stop the fight, causing Tan to faint on the spot. Tang called the "110" number to report the incident, and after being arrested, he truthfully confessed the incident. According to the appraisal, the victim Tan's injury was a second-degree severe injury.) |
| **Output2** | [刑期区间]B[eoa] ([SENTENCE] B [eoa]) |

Figure 6: Examples of the Sentence Prediction Task

**An Example of Capability Item $C_{02}$**

| | |
|---|---|
| **Instruction** | 请对以下给定文本完成结构化文本抽取任务，按照以下结构生成结构化文本1.罪名:xxx;\n2.犯罪行为：xxx\n3.法定刑罚细节：xxx
(Complete the structured text extraction task for the following text and generate structured text based on the following structure: 1. Crime: xxx;\n2. Crime: xxx\n3. Penalty details: xxx\n4.xxx) |
| **Input** | 第二百三十四条【故意伤害罪】
(Article 234 [Intentional Assault])
故意伤害他人身体的，处三年以下有期徒刑、拘役或者管制。犯前款罪，致人重伤的，处三年以上十年以下有期徒刑；致人死亡或者以特别残忍手段致人重伤造成严重残疾的，处十年以上有期徒刑、无期徒刑或者死刑。本法另有规定的，依照规定。
(Whoever intentionally causes bodily injury to another person shall be sentenced to fixed-term imprisonment of not more than three years, criminal detention or public surveillance. Whoever commits the crime mentioned in the preceding paragraph and causes serious injury to another person shall be sentenced to fixed-term imprisonment of not less than three years but not more than 10 years; Those who cause death to others or use especially cruel means to cause serious injury to others resulting in serious disability shall be sentenced to not less than 10 years of fixed-term imprisonment, life imprisonment, or death. Where this Law provides otherwise, such provisions shall prevail.) |
| **Output** | 1.罪名：故意伤害罪；(Charge: crime of intentional injury)
2.犯罪行为：(Offences)
- 故意伤害他人身体；(Intentional infliction of bodily harm)
- 致人重伤；(Causing serious injury;)
- 致人死亡或者以特别残忍手段致人重伤造成严重残疾；
(Causing death or serious injury resulting in serious disability by particularly cruel means)
3.法定刑罚细节：(Details of Statutory Penalties)
- 故意伤害他人身体：处三年以下有期徒刑、拘役或者管制；
(Intentional infliction of bodily injury: imprisonment, detention or restraint for not more than three years)
- 致人重伤：处三年以上十年以下有期徒刑；
(Causing serious bodily injury: imprisonment for not less than three years and not more than 10 years)
- 致人死亡或者以特别残忍手段致人重伤造成严重残疾处十年以上有期徒刑、无期徒刑或者死刑；
(Causing death or serious injury to a person by particularly cruel means, resulting in serious disability: imprisonment of not less than 10 years, life imprisonment or death)
- 本法另有规定的：依照规定。
(Where this Law provides otherwise: in accordance with the provisions) |

Figure 7: An example of Capability Item 1

**An Example of Capability Item $C_{12}$**

| | |
|---|---|
| **Instruction** | 请从输入中抽取给定范围内的实体，以[实体类型]:[实体内容]的格式输出。实体类型的范围如下：被告人，被害人，犯罪行为，鉴定结果，伤害程度。
(Extract entities in the specified range from the input and output them in the format [Entity type]:[Entity content]. The range of entity types is as follows: defendant, victim, criminal act, result of identification, degree of injury) |
| **Input** | 包头稀土高新技术产业开发区人民检察院指控，2014年7月11日8时许，在包头市昆都仑区二旅社长途汽车站售票厅门前，被告人王某甲与被害人王某乙因抢客发生口角互相推搡，王某甲用右拳打在王某乙左侧脸部腮帮处，至王某乙倒地后头部触地出血并昏迷。王某乙被120急救车送往包钢三医院救治，并于当日下午转至包头医学院第一附属医院，诊断为颅骨骨折、头皮挫裂伤等。2014年7月13日包头医学院第一附属医院对王某乙进行了头部开颅手术，取出颅内积血。后经包头市公安司法鉴定中心鉴定，王某乙所受伤损程度为重伤二级。
(The People's Procuratorate of Baotou Rare Earth Hi-Tech Industrial Development Zone alleged that at about 8 o'clock on July 11, 2014, in front of the ticket office of No. 2 Hotel Long-distance Bus Station in Kundulun District, Baotou City, the defendant A and the victim B had a quarrel and pushed each other for snatching passengers. A with his right fist in the left side of the face B cheek, B fell to the ground after the head touch bleeding and coma. B was sent to the third hospital of Baotou Steel by 120 ambulance, and transferred to the first affiliated hospital of Baotou Medical College in the afternoon of the same day, diagnosed with skull fractures, scalp contusion and laceration. On July 13, 2014, the First Affiliated Hospital of Baotou Medical College performed a craniotomy on B's head to remove intracranial blood. After identification by the Baotou City Public Security Judicial Appraisal Center, B was injured to the extent of two serious injuries.) |
| **Output** | 被告人:王某甲 (Defendant: Wang A)
被害人:王某乙 (Victim: Wang B)
犯罪行为:王某甲用右拳打在王某乙左侧脸部腮帮处，至王某乙倒地后头部触地出血并昏迷
(Crime: Wang A with his right fist wang B in the left side of the face cheek, to wang B fell to the ground after the head touch bleeding and coma)
鉴定结果:通过诊断为颅骨骨折、头皮挫裂伤等。在包头医学院第一附属医院进行了头部开颅手术，取出颅内积血
(Result of identification: Diagnosed as skull fracture, scalp contusion, etc. The first affiliated hospital in baotou medical school for a head craniotomy, remove intracranial hemorrhage)
伤害程度:重伤二级 (Injury: Level 2) |

Figure 8: An example of Capability Item 2

---

**An Example of Capability Item $C_{13}$**

| | |
|---|---|
| **Instruction** | 根据下列输入，判断出事件中受害人的伤害程度，从A,B,C三个受伤程度选项(A:轻伤; B:重伤; C:死亡或严重残疾)中选出一个最合适的受伤程度，必须给出一个选项。
(Based on the following inputs, determine the degree of injury of the victim in the incident. Choose from A, B, and C. (A: minor injury; B: serious injury; C: death or serious disability) To select the most appropriate degree of injury, an option must be given.) |
| **Input** | 犯罪行为:李某某朝董某扔棋子并向其冲过来，董某在后退过程中仰面摔倒，尚未起身时被李某某上前脚踹右侧肋部，致董某右侧第6、7肋骨骨皮质不连续
(Criminal behavior: Li threw chessmen at Dong and rushed towards him. Dong fell on his back in the process of retreating. Before getting up, he was kicked on the right rib by Li's front foot, causing discontinuity of the bone cortex of the 6th and 7th ribs on the right side of Dong.) |
| **Output** | A |

Figure 9: An example of Capability Item 3

---

**An Example of Capability Item $C_{22}$**

| | |
|---|---|
| **Instruction** | 请对输入的案例信息中的张某乙的判决答案进行简要解释。
(Briefly explain the answer to the question of Zhang B in the entered case information.)
刑期区间：A:三年以下有期徒刑、拘役或者管制；B:三年以上十年以下有期徒刑；C:十年以上有期徒刑、无期徒刑或死刑。
(Sentence Range: A: Up to three years of fixed-term imprisonment, criminal detention, or control; B: Imprisonment for not less than three years but not more than ten years; C: More than 10 years imprisonment, life imprisonment or death penalty.) |
| **Input** | 案例信息-上海市宝山区人民检察院指控，2015年10月14日21时30分许，被告人张某乙在上海市宝山区沪太路龙丰木材市场Q座门口处，就其承包工程事宜与发包人陈某发生争执引发肢体冲突，期间，张某乙造成陈某左第五掌骨完全性骨折，经鉴定，构成轻伤二级。审理中，被告人赔偿被害人经济损失人民币6万元，取得谅解。
(Case Information - The People's Procuratorate of Baoshan District, Shanghai alleges that at about 21:30 on October 14, 2015, the defendant Zhang B was at the entrance of Block Q,Longfeng Timber Market, Hutai Road, Baoshan District, Shanghai, On its contract with the contract matters arising from a dispute with the employer Chen physical conflict, during which Zhang B caused Chen left the fifth metacarpal bone fracture, identified, constitute a minor injury two. During the trial, the defendant compensated the victim for economic losses of 60,000 yuan and reached an understanding.)
判决答案-[刑期区间]A[eoa] (Sentencing Answer - [SENTENCE] A[eoa]) |
| **Output** | 根据您提供的案例信息及相关法条，张某乙因为与发包人陈某发生争执后引发的肢体冲突，致使陈某受到身体伤害，经鉴定为轻伤二级。根据《中华人民共和国刑法》第二百三十四条规定，故意伤害他人身体的若导致的后果不构成重伤，可处三年以下有期徒刑、拘役或者管制。在审理过程中，张某乙已经赔偿被害人经济损失，并且取得了被害人的谅解。因此，根据这些情况，张某乙可能会被判处三年以下的有期徒刑、拘役或者管制。
(According to the case information you provided and relevant laws, Zhang B caused physical conflict with the employer Chen after a dispute, resulting in physical injury to Chen, which was identified as a minor injury of the second degree. Article 234 of the People's Republic of China Criminal Law stipulates that anyone who intentionally injures another is liable to imprisonment. During the trial, Zhang B has compensated the victim for economic losses and obtained the victim's understanding. Therefore, based on these circumstances, Zhang may be sentenced to a fixed-term imprisonment of less than three years, criminal detention or control.) |

Figure 10: An example of Capability Item 4

---

**Prompt Template for Zero-shot CoT in Law Domain.**

**# Role:** 你是一名法官，请完成用户指定的任务。
(**# Role:** You are a judge. Please complete the tasks specified by the user.)

**任务描述**：根据下列输入，对被告人李某某进行刑期判决。
要求：从A,B,C三个刑期区间选项(A:三年以下有期徒刑、拘役或者管制；B:三年以上十年以下有期徒刑；C:十年以上有期徒刑、无期徒刑或死刑)中选出一个最合适的判决刑期区间，请将答案填在刑期区间与[eoa]之间，必须给出一个选项，请严格按要求的输出格式输出答案，不要输出任何无关内容或者解释。例如[刑期区间]C[eoa]。
(**Task description:** According to the following input, sentence the defendant Li.
Requirements: Select the most appropriate sentence interval from the three sentence interval options A, B, and C (A: imprisoment of less than three years, detention or control; B: imprisoment of more than three years but less than ten years; C: imprisoment of morethan ten years, life imprisonment or death penalty). Please fill in the answer between [SENTENCE] and [eoa]. You must give an option. Pleaseoutput the answer strictly in the required output format. Do not output any irrelevant content or explanation. For example, [SENTENCE] C[eoa].)

**输入 (Input):** [Please Put Your Input Here.]

**输出 (Output):**

Figure 11: The Prompt Template for Zero-shot CoT in Law Domain.

**Prompt Template for Few-shot CoT in Law Domain.**

**# Role:** 你是一名法官，请理解<参考法条>的基础上，完成用户指定的任务。
(**# Role:** You are a judge. Please understand the <reference law> and complete the tasks specified by the user.)

**# Demonstration:**
**[Please Put Your Demonstrations Here.]**

**任务描述**：根据下列输入，对被告人李某某进行刑期判决。
要求：从A,B,C三个刑期区间选项(A:三年以下有期徒刑、拘役或者管制；B:三年以上十年以下有期徒刑；C:十年以上有期徒刑、无期徒刑或死刑)中选出一个最合适的判决刑期区间，请将答案填在刑期区间与[eoa]之间，必须给出一个选项，请严格按要求的输出格式输出答案，不要输出任何无关内容或者解释。例如刑期区间]C[eoa]。
(**Task description:** According to the following input, sentence the defendant Li.
Requirements: Select the most appropriate sentence interval from the three sentence interval options A, B, and C (A: imprisoment of less than three years, detention or control; B: imprisoment of more than three years but less than ten years; C: imprisoment of morethan ten years, life imprisonment or death penalty). Please fill in the answer between [SENTENCE] and [eoa]. You must give an option. Pleaseoutput the answer strictly in the required output format. Do not output any irrelevant content or explanation. For example, [SENTENCE] C[eoa].)

**输入 (Input): [Please Put Your Input Here.]**

**输出 (Output):**

Figure 12: The Prompt Template for Few-shot CoT in Law Domain.

**Prompt Template for Re-Task (Lite) in Law Domain.**

**# Role:** 你是一名法官，请完成用户指定的任务。
(**# Role:** You are a judge. Please complete the tasks specified by the user.)

**# 参考法条**：
(**# Reference law**)
**[Please Put the Knowledge Here.]**

**# Demonstration:**
**[Please Put Your Capability Items for the Overall Task Here.]**

**任务描述**：根据下列输入，对被告人李某某进行刑期判决。
要求：从A,B,C三个刑期区间选项(A:三年以下有期徒刑、拘役或者管制；B:三年以上十年以下有期徒刑；C:十年以上有期徒刑、无期徒刑或死刑)中选出一个最合适的判决刑期区间，请将答案填在刑期区间与[eoa]之间，必须给出一个选项，请严格按要求的输出格式输出答案，不要输出任何无关内容或者解释。例如刑期区间]C[eoa]。
(**Task description:** According to the following input, sentence the defendant Li.
Requirements: Select the most appropriate sentence interval from the three sentence interval options A, B, and C (A: imprisoment of less than three years, detention or control; B: imprisoment of more than three years but less than ten years; C: imprisoment of morethan ten years, life imprisonment or death penalty). Please fill in the answer between [SENTENCE] and [eoa]. You must give an option. Pleaseoutput the answer strictly in the required output format. Do not output any irrelevant content or explanation. For example, [SENTENCE] C[eoa].)

**输入 (Input): [Please Put Your Input Here.]**

**输出 (Output):**

Figure 13: The Prompt Template for Re-TASK (Lite) in Law Domain.

**Prompt Template for Re-Task (Full) in Law Domain.**

**# Role:** 你是一名法官，请完成用户指定的任务。
(**# Role:** You are a judge. Please complete the tasks specified by the user.)

**# 参考法条**：
(**# Reference law**)
**[Please Put the Knowledge Here.]**

**# Demonstration:**
**[Please Put Your Capability Items for the Subtasks Here.]**

**[Please Put Your Capability Items for the Overall Task Here.]**

**任务描述**：根据下列输入，对被告人李某某进行刑期判决。
要求：从A,B,C三个刑期区间选项(A:三年以下有期徒刑、拘役或者管制；B:三年以上十年以下有期徒刑；C:十年以上有期徒刑、无期徒刑或死刑)中选出一个最合适的判决刑期区间，请将答案填在刑期区间]与[eoa]之间，必须给出一个选项，请严格按要求的输出格式输出答案，不要输出任何无关内容或者解释。例如刑期区间]C[eoa]。
(**Task description:** According to the following input, sentence the defendant Li.
Requirements: Select the most appropriate sentence interval from the three sentence interval options A, B, and C (A: imprisoment of less than three years, detention or control; B: imprisonment of more than three years but less than ten years; C: imprisonment of morethan ten years, life imprisonment or death penalty). Please fill in the answer between [SENTENCE] and [eoa]. You must give an option. Pleaseoutput the answer strictly in the required output format. Do not output any irrelevant content or explanation. For example, [SENTENCE] C[eoa].)

**输入 (Input): [Please Put Your Input Here.]**

**输出 (Output):**

Figure 14: The Prompt Template for Re-TASK (Full) in Law Domain.

Table 15: The prompt template of one-shot CoT on MMLU-Math dataset.

**Prompt template of One-shot CoT**

**Input:**
# Role:
You are an expert in the field of Math. Complete the task provided by the user.

# Demonstration:
Question:
[Please Put Your Question of Demonstration Here]
Options:
[Please Put Your Options of Demonstration Here]
Rationale:
[Please Put Your Rationale of Demonstration Here]
Correct: [Please Put Your Final Choice of Demonstration Here]

# Task Description:
According to the following question and options, refer to <# Demonstration>, select one correct option. Please think step by step and follow the answer format in <# Demonstration>, i.e., output the brief chain-of-thought beginning with 'Rationale: ' within 500 characters, and the final choice option beginning with 'Correct: ' from 'A' to 'D'.
Question:
[Please Put Your Question Here]
Options:
[Please Put Your Options Here]

**Prompt For Standard Prompting + Capability Item 3 + CoT**

**System Prompt**
**# Role:** 你是一名法官，请理解<参考法条>的基础上，完成用户指定的任务。
(# Role: You are a judge. Please understand the <reference law> and complete the tasks specified by the user.)

**# 参考法条：**
**(# Reference law)**
第二百三十四条【故意伤害罪】故意伤害他人身体的，处三年以下有期徒刑、拘役或者管制。犯前款罪，致人重伤的，处三年以上十年以下有期徒刑；致人死亡或者以特别残忍手段致人重伤造成严重残疾的，处十年以上有期徒刑、无期徒刑或者死刑。
(Article 234 [Intentional injury] Anyone who intentionally injures another person shall be sentenced to fixed -term imprisonment of not more than three years, criminal detention or public surveillance. Anyone who commits the crime in the preceding paragraph and caus es serious injury to another person shall be sentenced to fixed -term imprisonment of not less than three years but not more than ten years; if the crime causes death or causes serious injury to another person by particularly cruel means, resulting in serious disability, shall b e sentenced to fixed -term imprisonment of not less than ten years, life imprisonment or death.)

**# Demonstration:**
**任务描述：**根据下列输入，判断出事件中受害人的伤害程度，从A,B,C三个受伤程度选项(A:轻伤; B:重伤; C:死亡或严重残疾)中选出一个最合适的受伤程度，请将答案填在[受伤程度]与[eoa]之间，例如[受伤程度]C[eoa]。必须选出一个选项，并严格按照要求的输出格式输出答案。
(**Task description:** Based on the following input, determine the injury level of the victim in the incident, and select the most appropriate injur y level from the three injury level options A, B, and C (A: minor injury; B: serious injury; C: death or severe disability). Pl ease fill in the answer between [INJURY] and [eoa], for example, [INJURY]C[eoa]. Give one option and output the answer strictly in the required outpu t format.)

**输入：**犯罪行为-厮打中，李某拉住曹某某的胳膊，曹某某持刀将曹某某腹部捅伤。经鉴定曹某某的伤情构成重伤。
(**Input:** Criminal behavior - During the fight, Li grabbed Cao's arm, and Cao stabbed Cao in the abdomen with a knife. Cao's injuries were identified as serious injuries.)

**输出：**[受伤程度]B[eoa]
(**Output:**[INJURY]B[eoa])

**任务描述：**根据下列输入，对被告人李某1进行刑期判决。
要求：从A,B,C三个刑期区间选项(A:三年以下有期徒刑、拘役或者管制; B:三年以上十年以下有期徒刑; C:十年以上有期徒刑、无期徒刑或死刑)中选出一个最合适的判决刑期区间，请将答案填在[刑期区间]与[eoa]之间，例如[刑期区间]C[eoa]。
(**Task description:** According to the following input, sentence the defendant Wu.
Requirements: Select the most appropriate sentence interval from the three sentence interval options A, B, and C (A: imprison ment of less than three years, detention or control; B: imprisonment of more than three years but less than ten years; C: imprisonment of more than ten years, life imprisonment or death penalty). Please fill in the answer between [SENTENCE] and [eoa]. For example, [SENTENCE]C[eoa].)

**输入：**2016年11月9日上午10时许，陈某1、李某1、李某2等人带着挖掘机在东源县上莞镇新南村的一座山地上进行修路施工，在施工过程中，新南村南兴小组村民陈某3、陈某4等人去到现场，因土地纠纷问题，陈某3与陈某1产生争执，继而双方发生打斗，双方人员在劝架过程中发生肢体接触，李某1用右手握拳打向陈某4的嘴巴，并致使陈某4门牙脱落。经东源县公安司法鉴定中心鉴定：陈某4身体的损伤已构成轻伤二级。2016年11月27日，被告人李某1家属与被害人陈某4达成调解协议书，李某1一次性付清被害人医疗费22000元且已履行完毕，陈某4出具谅解书，对李某1的行为予以谅解，自愿不再追究李某1的刑事责任。
(**Input:** At about 10 am on November 9, 2016, Chen 1, Li 1, Li 2 and others took an excavator to carry out road construction on a mountain in Xinnan Village, Shangguan Town, Dongyuan County. During the construction process, Chen 3, Chen 4 and other villagers from the Nanxing Group of Xinnan Village went to the scene. Due to land disputes, Chen 3 and Chen 1 had a dispute, and then the two sides fought. During the process of persuading the two sides, physical contact occurred. Li 1 punched Chen 4 in the mouth with his right fist, causing Chen 4's front teeth to fall out. According to the appraisal of the Dongyuan County Public Security Judicial Appraisal Center: Chen 4's physical injuries have constituted a second -degree minor injury. On November 27, 2016, the family of the defendant Li 1 and the victim Chen 4 reached a mediation agreement. Li 1 paid the victim's medical expenses of 22,000 yuan in one lump sum and has fulfilled it. Chen 4 issued a lette r of understanding, forgiving Li 1's behavior and voluntarily no longer pursuing Li 1's criminal responsibility.)

**输出：**根据所提供的案例信息，李某1因土地纠纷在争执中用拳头打到陈某4并致使其门牙脱落，经鉴定陈某4的伤势已构成轻伤二级。而按照参考法条的规定，这种故意伤害他人身体的行为属于故意伤害罪，刑期应在三年以下有期徒刑、拘役或者管制范围内，即刑期区间A。另外，李某1已通过支付医疗费并得到了陈某4的谅解，陈某4自愿不再追究其刑事责任，这些因素也可能影响到最终的判决结果。所以答案是[刑期区间]A[eoa]。
(**Output:** According to the case information provided, Li No. 1 punched Chen No. 4 in a dispute over land and caused his front teeth to fall out. Chen No. 4's injuries were identified as second -degree minor injuries. According to the reference law, this act of intentionally injuring another person's body is a crime of intentional injury, and the sentence should be within the range of imprisonment of less than thre e years, detention or control, that is, the sentence range A. In addition, Li No. 1 has paid medical expenses and obtained Chen No. 4's forgiven ess, and Chen No. 4 voluntarily no longer pursues his criminal responsibility. These factors may also affect the final judgment. So the answer is [SENTENCE]A[eoa].)

**User Prompt**
**任务描述：**根据下列输入，对被告人李某某进行刑期判决。
要求：从A,B,C三个刑期区间选项(A:三年以下有期徒刑、拘役或者管制; B:三年以上十年以下有期徒刑; C:十年以上有期徒刑、无期徒刑或死刑)中选出一个最合适的判决刑期区间，请将答案填在[刑期区间]与[eoa]之间，必须给出一个选项，请严格按要求的输出格式输出答案，不要输出任何无关内容或者解释。例如[刑期区间]C[eoa]。
(**Task description:** According to the following input, sentence the defendant Li.
Requirements: Select the most appropriate sentence interval from the three sentence interval options A, B, and C (A: imprison ment of less than three years, detention or control; B: imprisonment of more than three years but less than ten years; C: imprisonment of more than ten years, life imprisonment or death penalty). Please fill in the answer between [SENTENCE] and [eoa]. You must give an option. Please output the answer strictly in the required output format. Do not output any irrelevant content or explanation. For example, [SENTENCE] C [eoa].)

**输入：**烟台市烟台经济技术开发区人民检察院指控：被告人李某某于2017年3月9日12时许，在烟台开发区大家家某某公司北门大院内，因工友王某某指使自己扔废料而起争执，后对王某某拳打脚踢。期间，被告人李某某按住王某某头部撞向旁边铁皮柜，致王某某额头受伤。经鉴定，王某某面部皮肤裂伤属轻伤一级。案发后，被告人李某某于2017年3月10日主动投案。就上述指控的事实，公诉机关向法庭提交了相应的证据予以证实。
(**Input:** Yantai Economic and Technological Development Zone People's Procuratorate of Yantai City accused: At about 12:00 on March 9, 2017, the defendant Li had a dispute with his co -worker Wang in the north gate courtyard of Dajijia Company in Yantai Development Zone because Wang instructed him to throw away waste, and then punched and kicked Wang. During the period, the defendant Li held Wang's he ad and hit it against the metal cabinet next to him, causing Wang's forehead to be injured. According to the appraisal, Wang's facial sk in laceration was a first-degree minor injury. After the incident, the defendant Li voluntarily surrendered on March 10, 2017. Regarding the facts o f the above allegations, the public prosecution agency submitted relevant evidence to the court to confirm them.)

**输出：**
(**Output:**)

Figure 15: The Prompting for Re-TASK (Full) using Capability Item $C_{13}$

Table 16: The prompt template of Re-TASK prompt on MMLU-Math dataset.

| Prompt template of Re-TASK |
| --- |

**Input:**
# Role:
You are an expert in the field of Math. Complete the task provided by the user.

# Knowledge:
[Please Put Your Knowledge Here]

# Demonstration:
Question:
[Please Put Your Question of Demonstration Here]
Options:
[Please Put Your Options of Demonstration Here]
Rationale:
[Please Put Your Rationale of Demonstration Here]
Correct: [Please Put Your Final Choice of Demonstration Here]

# Task Description:
According to the following question and options, refer to <# Knowledge> and <# Demonstration>, select one correct option. Please think step by step and follow the answer format in <# Demonstration>, i.e., output the brief chain-of-thought beginning with 'Rationale: ' within 500 characters, and the final choice option beginning with 'Correct: ' from 'A' to 'D'.
Question:
[Please Put Your Question Here]
Options:
[Please Put Your Options Here]

Table 17: The prompt template of knowledge generation.

| Prompt template of Knowledge Generation |
| --- |

# Role:
You are an expert in the field of Math. Complete the task provided by the user.

# Demonstration:
## Question: The hypotenuse of a right triangle measures 10 inches and one angle is $45^{\circ}$. What is the number of square inches in the area of the triangle?
## Knowledge: The area of a right triangle is given by A = (1/2) * base * height.

# Task Description:
Given the question, please just generate the formula or other knowledge related to the question as brief as possible, like the <# Demonstration>. Just output the one related formula or other knowledge, DO NOT output any other characters.
## Question:
[Please Put Your Question Here]
## Knowledge:

Table 18: The prompt template of capability item generation.

**Prompt template of Capability Item Generation**

\# Role:
You are an expert in the field of Math. Complete the task provided by the user.

\# Demonstration:
{
    "question":
        "At a certain factory, 10 percent of the staplers produced on Monday were defective and 2 percent of the non-defective staplers were rejected by mistake. If 72 of the non-defective staplers were rejected, what was the number of staplers produced that day?"
    "options": [
        "A. 4,000",
        "B. 4,200",
        "C. 4,500",
        "D. 4,800"
    ]
    "rationale":
        "Step 1. We're told that 10% of staplers in a factory are defective. \n Step 2. X = Total staplers, 0.1X = defective staplers, 0.9X = normal staplers. \n Step 3. We're told that 2% of the normal staplers were rejected by mistake and that this = 72 staplers. \n Step 4. 0.9X(0.02) = 72, 0.018X = 72, 18X = 72,000, X = 4,000.",
    "correct": "A"
}

\# Task Description:
I will give you a piece of knowledge text, please help me generate a four-choice question which is one deduction application of this knowledge, including the question, options, rationale and correct answer.
The knowledge is [Please Put Your Knowledge Here].
The answer is required to follow the **json** format in <# Demonstration>, as:

{
    "question": Content of the question,
    "options": list of four options,
    "rationale": Content of the chain-of-thought with each step starting with 'Step x. ', which is limited to 400 characters,
    "correct": the single choice character of correct answer
}
You must and can only generate one deduction example of the given knowledge in the above json format. No extra characters are allowed.

---

**Prompt Template for Zero-shot CoT on FinanceIQ Dataset.**

**# Role:** 你是一名经济领域的专家，请完成指定的任务。
(**# Role:** You are an expert in the field of economics, please complete the tasks specified by the user.)

**任务描述**：现在根据下列输入的问题，从4个选项中选择一个正确的选项，请将答案填在[选项]与[选项]之间，例如[选项]C[选项]，必须给出一个选项。
(**Task description:** Now, based on the following input question, select the correct option from the 4 options. Please fill in the answer between [options] and [options], for example [option] C [option], one option must be given.)

**问题 (Question): [Please Put Your Questions Here.]**
**选项 (Options):  [Please Put Your Options Here.]**

**答案 (Answer):**

Figure 16: The Prompt Template for Zero-shot CoT on FinanceIQ Dataset.

---

**Prompt Template for Few-shot CoT using random Demo CoT on FinanceIQ Dataset.**

---

**# Role:** 你是一名经济领域的专家，请完成指定的任务。
(**# Role:** You are an expert in the field of economics, please complete the tasks specified by the user.)

**# Demonstration:**
**[Please Put the random Demonstrations Here.]**

**# Initialization**
作为<Role>，你可以参考<Demonstration>完成用户给定的任务。
(As a<Role>, you can refer to<Demonstration>to complete the tasks given by the user.)

**任务描述**：现在根据下列输入的问题，从4个选项中选择一个正确的选项，请将答案填在[选项]与[选项]之间，例如[选项]C[选项]，必须给出一个选项。
(**Task description:** Now, based on the following input question, select the correct option from the 4 options. Please fill in the answer between [options] and [options], for example [option] C [option], one option must be given.)

**问题 (Question):** **[Please Put Your Questions Here.]**
**选项 (Options):** **[Please Put Your Options Here.]**

**答案 (Answer):**

---

Figure 17: The Prompt Template for Few-shot CoT on FinanceIQ Dataset.

---

**Prompt Template for Re-Task (Lite) on FinanceIQ Dataset.**

---

**# Role:** 你是一名金融领域的专家，请通过对<Knowledge Application>的学习加深对响应的<Knowledge Recall>的应用能力，完成用户指定的任务。
 (**# Role:** You are an expert in the field of economics, please deepen your application ability of<Knowledge Recall>through learning<Knowledge Application> and complete the tasks specified by the user.)

**# Capability Items for Overall Task:**
**## Knowledge Recall**
**[Please Put the Knowledge Recall of the overall Task Here.]**

**## Knowledge Application**
**[Please Put the Knowledge Application of the overall Task Here.]**

**# Initialization**
作为<Role>，你可以参考<Demonstration>完成用户给定的任务。
(As a<Role>, you can refer to<Demonstration>to complete the tasks given by the user.)

**任务描述**：现在根据下列输入的问题，从4个选项中选择一个正确的选项，请将答案填在[选项]与[选项]之间，例如[选项]C[选项]，必须给出一个选项。
(**Task description:** Now, based on the following input question, select the correct option from the 4 options. Please fill in the answer between [options] and [options], for example [option] C [option], one option must be given.)

**问题 (Question):** **[Please Put Your Questions Here.]**
**选项 (Options):** **[Please Put Your Options Here.]**

**答案 (Answer):**

---

Figure 18: The Prompt Template for Re-TASK (Lite) on FinanceIQ Dataset.

---

**Prompt Template for Re-Task (Full) on FinanceIQ Dataset.**

---

**# Role:** 你是一名金融领域的专家，请通过<Knowledge Example 1>的学习加深对响应的<Knowledge Recall 1>的理解能力，对通过对
<Knowledge Application 2>的学习加深对响应的<Knowledge Recall 2>的应用能力，完成用户指定的任务。
 (**# Role:** You are an expert in the field of economics, please deepen your understanding of <Knowledge Recall 1> through learning <Knowledge Example 1> and deepen your application ability of <Knowledge Recall> through learning <Knowledge Application> and complete the tasks specified by the user.)

**# Capability Items for Subtasks:**
**## Knowledge Recall 1**
**[Please Put the Knowledge Recall 1 of Subtasks Here.]**

**## Knowledge Example 1**
**[Please Put the Knowledge Example 1 of Subtasks Here.]**

**## Knowledge Recall 2**
**[Please Put the Knowledge Recall 2 of the Subtasks Here.]**

**## Knowledge Application 2**
**[Please Put the Knowledge Application 2 of the Subtasks Here.]**

**# Capability Items for Overall Task:**
**## Knowledge Recall**
**[Please Put the Knowledge Recall of the overall Task Here.]**

**## Knowledge Application**
**[Please Put the Knowledge Application of the overall Task Here.]**

**# Initialization**
作为<Role>，你可以参考<Demonstration>完成用户给定的任务。
(As a<Role>, you can refer to<Demonstration>to complete the tasks given by the user.)

**任务描述**：现在根据下列输入的问题，从4个选项中选择一个正确的选项，请将答案填在[选项]与[选项]之间，例如[选项]C[选项]，必须给出一个选项。
(**Task description:** Now, based on the following input question, select the correct option from the 4 options. Please fill in the answer between [options] and [options], for example [option] C [option], one option must be given.)

**问题 (Question):** [Please Put Your Questions Here.]
**选项 (Options):** [Please Put Your Options Here.]

**答案 (Answer):**

---

Figure 19: The Prompt Template for Re-TASK (Full) on FinanceIQ Dataset.

