# OpenReview forum: "Re-TASK: Revisiting LLM Tasks from Capability, Skill, and Knowledge Perspectives"
_ICLR.cc/2025/Conference — Submitted to ICLR 2025_

### Official Review · Reviewer_esD1 · 2024-10-26

**Soundness:** 3
**Presentation:** 2
**Contribution:** 1
**Rating:** 3
**Confidence:** 3

**Summary:**

The paper proposes a prompting strategy inspired by education theories on knowledge. Built on top of CoT (chain of thoughts), the ReTASK strategy includes relevant "capabilities items" in the prompts to help LLMs break down the task to subtasks, inject relevant knowledge, and demonstrate how the knowledge can be applied.

The capability items for a given task can be manually curated or generated by strong LLMs.

The experiments show prompts containing relevant "capability items" substantial quality gains over CoT on benchmarks including Chinese legal sentencing, FinanceIQ, and MMLU high school math.

**Strengths:**

[Originality] It's a novel idea to apply Bloom’s Taxonomy and Knowledge Space Theory to LLM prompting.
[Significance] LM prompting strategies have large impacts on the performance and are an important topic to study.

**Weaknesses:**

[Clarity] The connections to the educational theories seem strenuous. The term "capability item" also seems unnecessarily abstract. A clearer message would be "hints generated by a large LM improve problem solving of smaller LMs". The paper can also benefit from providing a detailed description about how high quality hints ("capability items") are generated with large LMs, e.g., what's the prompting strategy used?

[Significance] It is not surprising that hints from high quality LLMs of ~70B params can improve the performance of LLMs of size ~7B. The gains on ~30B models seem to be much more modest.

[Significance] If we have a general strategy to generate high quality hints, the experiments should include many more benchmarks, e.g., all of MMLU (instead of limiting to high school math), GPQA, MATH, HumanEval-Python, Arena-Hard, to name a few. A rating from the LMSys Chatbot Leaderboard would also be more convincing.

**Questions:**

In the prompt templates show in the appendix (Figure 19 and 20), there are placeholders such as "[Please Put the Knowledge Recall 1 of Subtasks Here.]" and "[Please Put the Knowledge Application of the overall Task Here.]"

Are these placeholders replaced by paragraphs containing the relevant knowledge in the LM input prompts? If so, what's the process used to retrieve or generate the relevant paragraphs?

---

### Official Review · Reviewer_RMwe · 2024-11-03

**Soundness:** 2
**Presentation:** 2
**Contribution:** 2
**Rating:** 3
**Confidence:** 3

**Summary:**

This paper proposes an LLM prompting framework called Re-TASK, which utilizes ideas from education and cognitive science to improve prompting efficacy. Specifically, the framework leverages Bloom's Taxonomy and defines concepts like tasks, capability items, knowledge, and skills. It employs a more structured approach to complete a given task through a series of instructions/subtasks. Meanwhile, the order of these subtasks is guided by principles borrowed from the Knowledge Space Theory. The paper shows that the proposed framework is effective by testing it with some Chinese LLMs on law/finance-related tasks and testing it with llama and mistral on a math dataset.

**Strengths:**

1. Integrating ideas from education and cognitive science into improving LLM prompting is quite novel.

**Weaknesses:**

Main conerns:
1. The prompting framework requires significant manual effort and domain expertise---it can only be done by experts who can successfully identify the capacity items, decompose the task, and apply all structured prompting techniques. The bar for using it seems too high.
2. Most evaluation uses custom datasets, which makes it hard to compare the proposed framework on more general tasks with a wider range of related methods. Also, the fact that two of the three evaluation domains are limited to datasets and LLMs in Chinese makes the results less strong.
3. The paper only considers CoT as baselines. Are there other methods applicable to the datasets?
4. There's a line of related research that prompts LLMs to generate relevant knowledge and then incorporate the knowledge into the question to improve the answer quality (see for example [1]). How's the proposed method related to and different from these works?
5. An important axis to evaluating prompting methods is whether the method can also help improve training. For instance, CoT can be used to reformulate training data, and by training with CoT, we can improve the LLM's ability in reasoning. However, the proposed Re-TASK framework seems too complicated and difficult to adapt for training.


Other minor issues:
1. To make the presentation clearer, there could be more introductions on the cognitive science related concepts.
2. Figure 2 can be moved to earlier (e.g., to the introduction section) to give readers a better sense of the prompting framework. It would also help improve the paper's clarity by accompanying Figur 2 with a concrete example. Currently the illustration is too high-level, abstract, and difficult for readers to understand. There could also be some notation definition in the caption, e.g., what does C_{ij} represent? Adding it could make it easier to understand as a standalone figure.
3. Section 4.4 seems misplaced. Moving it to the beginning of Section 4 might help improve presentation clarity.

Reference
[1] Generated Knowledge Prompting for Commonsense Reasoning. Liu, Jiacheng and Liu, Alisa and Lu, Ximing and Welleck, Sean and West, Peter and Bras, Ronan Le and Choi, Yejin and Hajishirzi, Hannaneh.

**Questions:**

See weaknesses

---

### Official Review · Reviewer_UU3p · 2024-11-05

**Soundness:** 2
**Presentation:** 2
**Contribution:** 2
**Rating:** 3
**Confidence:** 4

**Summary:**

This paper introduces the Re-TASK framework, which aims to enhance the performance of LLMs in domain-specific tasks by integrating domain-specific knowledge acquisition and skill adaptation. By combining the CoT approach with Re-TASK, the authors propose a novel prompting strategy that targets both capability and knowledge aspects within tasks. Their approach demonstrates performance improvements across three domains (law, finance, and mathematics) using open-source LLMs.

**Strengths:**

- The framework’s design is motivated by a learning theory, which adds a theoretical foundation to the methodology.
- The distinction between knowledge and skill acquisition is an interesting concept, reflected in the architecture and prompting strategy.
- The results are promising for domain-specific applications

**Weaknesses:**

- A primary limitation is the choice to limit experiments to open-source models. It’s not clear why the authors couldn’t include a proprietary model like GPT-4, as this would help validate how the framework performs across a wider range of LLMs.
The process of constructing capabilities seems relatively manual, which may restrict the framework's scalability to broader, less-defined tasks outside highly specialized domains.
- Constructing capabilities still seem fairly manual at this point, which might make it challenging to apply this framework to more general tasks. With specialized domains, it’s a bit easier to pre-define relevant knowledge, but for broader tasks, the approach could lose some of its practicality.
- Retrieving and injecting relevant knowledge has been explored in previous work in RAG. It would be helpful to clarify how this approach is distinct from those existing methods.
- Section 4.3 on automatically constructing capability items could use more detail, especially since this part feels critical for broader applications.
- While the framework shows significant gains in the law domain but not as much in two other domains. This might stem from manually obtained procedural knowledge. This suggests the framework might need predefined knowledge assumptions, which could limit its flexibility.

**Questions:**

- The term “skill adaptation” felt somewhat ambiguous, though Section 3.2 provided more clarity with its explanation of conceptual and procedural knowledge. Authors could consider adding a more concrete example to illustrate this concept clearly in the main text
- Providing a real-world example in the main paper would greatly help readers to understand the core components of the framework

---

### Meta-Review · Area_Chair_zSBj · 2024-12-21

**Metareview:**

The paper introduces the Re-TASK framework, combining CoT reasoning with structured knowledge injection and skill adaptation to enhance the performance of LLMs on domain-specific tasks. While the paper presents a novel application of educational theories like Bloom’s Taxonomy, reviewers raised concerns, including limited generalizability, reliance on custom datasets that hinder comparability, and insufficient exploration of related work on RAG. The authors didn't respond to the reviewers' comments during the rebuttal. I recommend reject this paper.

**Additional Comments On Reviewer Discussion:**

There was no reviewer discussion because the authors didn't reply to reviewers' comments.

---

### Decision · Program_Chairs · 2025-01-22

Reject